# Robust Distributed Estimation: Extending Gossip Algorithms to Ranking and Trimmed Means

**Anna van Elst**    **Igor Colin**    **Stephan Clémençon**
LTCI, Télécom Paris, Institut Polytechnique de Paris
`{anna.vanelst, igor.colin, stephan.clemencon}@telecom-paris.fr`

## Abstract

This paper addresses the problem of robust estimation in gossip algorithms over arbitrary communication graphs. Gossip algorithms are fully decentralized, relying only on local neighbor-to-neighbor communication, making them well-suited for situations where communication is constrained. A fundamental challenge in existing mean-based gossip algorithms is their vulnerability to malicious or corrupted nodes. In this paper, we show that an outlier-robust mean can be computed by globally estimating a robust statistic. More specifically, we propose a novel gossip algorithm for rank estimation, referred to as GORANK, and leverage it to design a gossip procedure dedicated to trimmed mean estimation, coined GOTRIM. In addition to a detailed description of the proposed methods, a key contribution of our work is a precise convergence analysis: we establish an $\mathcal{O}(1/t)$ rate for rank estimation and an $\mathcal{O}(1/t)$ rate for trimmed mean estimation, where by $t$ is meant the number of iterations. Moreover, we provide a breakdown point analysis of GOTRIM. We empirically validate our theoretical results through experiments on diverse network topologies, data distributions and contamination schemes.

## 1  Introduction

Distributed learning has gained significant attention in machine learning applications where data is naturally distributed across a network, either due to resource or communication constraints [7]. The rapid development of the Internet of Things has further amplified this trend, as the number of connected devices continues to grow, producing large volumes of data at the edge of the network. As a result, edge computing, where computation is performed closer to the data source, has emerged as a viable alternative to conventional cloud computing [6, 36]. By reducing reliance on centralized servers, edge computing offers several advantages, including lower energy consumption, improved data security, and reduced latency [6, 36]. In this context, several distributed learning frameworks, including Gossip Learning and Federated Learning, have been developed. Federated Learning depends on a central server for model aggregation [26], which introduces challenges such as communication bottlenecks and a single point of failure. In contrast, Gossip Learning provides a fully decentralized alternative where nodes communicate only with their nearby neighbors [18]. In addition, Gossip learning is particularly well-suited for situations where communication is constrained, such as in peer-to-peer or sensor networks.

One of the central challenges in distributed learning is robustness to corrupted nodes—scenarios where some nodes may contain contaminated data due to hardware faults, or even adversarial attacks [8, 39]. Consider a network of sensors deployed to monitor the temperature in a small region, as described in [5]. In idealized scenarios, sensor noise is typically modeled by a zero-mean Gaussian. However, a recent survey (see [1]) mentions that sensor networks are especially prone to outliers due to their reliance on imperfect sensing devices. Indeed, in practice, sensors can malfunction (e.g., become miscalibrated, get stuck at constant values, or report extreme values

39th Conference on Neural Information Processing Systems (NeurIPS 2025).

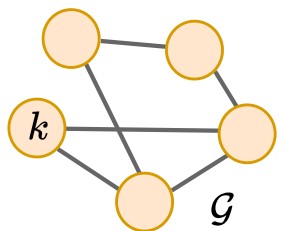 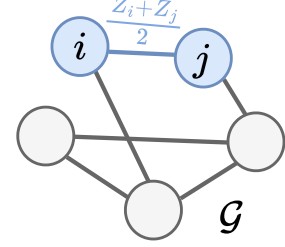 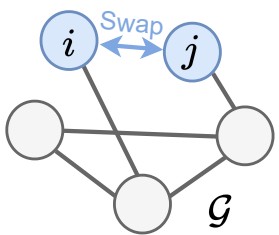

**1.** $\forall k \in V$: update rank $R_k$ and trimmed mean $Z_k$ estimates

**2.** Select $(i,j) \in E$ randomly and average $Z_i$ and $Z_j$

**3.** Swap observations $X_i$ and $X_j$

Figure 1: Illustration of a GOTRIM's iteration paired with GORANK on a communication graph $\mathcal{G} = (V, E)$. Initially, each node $k \in V$ stores the observation $X_k$.

due to environmental interference). These issues introduce outliers in the dataset which can corrupt the estimated temperature. Specifically, we consider a setting in which a fraction of the nodes hold outlying data [14]. In Federated Learning, significant progress has been made in designing robust aggregation methods based on statistics, such as the median and trimmed mean [2, 3, 39]. These statistics are known to be more robust to outliers than the standard mean [19]. However, ensuring robustness in Gossip Learning remains challenging. Unlike Federated Learning, where a central server can enforce robust aggregation rules, gossip algorithms inherently rely on local communication. To the best of our knowledge, approaches relying on globally estimating robust statistics remain largely unexplored in the gossip literature.

In this paper, we address the problem of robust estimation in gossip algorithms over arbitrary communication graphs. Specifically, we show that an outlier-robust mean can be computed by globally estimating a robust statistic. Ranks play a crucial role in the framework we develop here, as they allow us to identify outliers and compute robust statistics on the one hand [30], and because they can be obtained by means of pairwise comparisons on the other. This allows us to exploit the gossip approach proposed in [10] to compute pairwise averages ($U$-statistics) in a decentralized way. Note that both ranks and trimmed means are inherently global statistics, which makes them more challenging to estimate in a decentralized manner compared to statistics like averages, minima, or maxima that rely on local conservation principles.

Our main contributions are summarized as follows:

- We propose GORANK, a new gossip algorithm for rank estimation, and establish the first theoretical convergence rates for gossip-based ranking on arbitrary communication graphs, proving an $\mathcal{O}(1/t)$ convergence rate, where $t \geq 1$ denotes the number of iterations.
- We introduce GOTRIM, the first gossip algorithm for trimmed mean estimation which can be paired with any ranking algorithm. GOTRIM does not rely on strong graph topology assumptions—a common restriction in robust gossip algorithms. We prove a competitive convergence rate of order $\mathcal{O}(1/t)$ and provide a breakdown point analysis, demonstrating the robustness of its estimate.
- Finally, we conduct extensive experiments on various contaminated data distributions and network topologies. Numerical results empirically show that (1) GORANK consistently outperforms previous work in large and poorly connected communication graphs, and (2) GOTRIM effectively handles outliers—its estimate quickly improves on the naive mean and converges to the true trimmed mean.

This paper is organized as follows. Section 2 introduces the problem setup and reviews the related works. In Section 3, we present our new gossip algorithm for ranking, along with its convergence analysis and supporting numerical experiments. Section 4 introduces our novel gossip algorithm for trimmed mean estimation, together with a corresponding convergence analysis and experimental results. Finally, Section 5 explores potential extensions of our work. Due to space constraints, technical details, further discussions, and results are deferred to the Supplementary Material.

## 2 Background and Preliminaries

This section briefly introduces the concepts of decentralized learning, describes the problem studied and the framework for its analysis.

## 2.1 Problem Formulation and Framework

Here, we formulate the problem using a rigorous framework and introduce the necessary notations.

**Notation.** Let $n \geq 1$. We denote scalars by normal lowercase letters $x \in \mathbb{R}$, vectors (identified as column vectors) by boldface lowercase letters $\mathbf{x} \in \mathbb{R}^n$, and matrices by boldface uppercase letters $\mathbf{X} \in \mathbb{R}^{n \times n}$. The set $\{1, \ldots, n\}$ is denoted by $[n]$, $\mathbb{R}_n$'s canonical basis by $\{\mathbf{e}_k : k \in [n]\}$, the indicator function of any event $\mathcal{A}$ by $\mathbb{I}_{\mathcal{A}}$, the transpose of any matrix $\mathbf{M}$ by $\mathbf{M}^\top$ and the cardinality of any finite set $F$ by $|F|$. By $\mathbf{I}_n$ is meant the identity matrix in $\mathbb{R}^{n \times n}$, by $\mathbf{1}_n = (1, \ldots, 1)^\top$ the vector in $\mathbb{R}^n$ whose coordinates are all equal to one, by $\| \cdot \|$ the usual $\ell_2$ norm, by $\lfloor \cdot \rfloor$ the floor function, and by $\mathbf{A} \odot \mathbf{B}$ the Hadamard product of matrices $\mathbf{A}$ and $\mathbf{B}$. We model a network of size $n > 0$ as an undirected graph $\mathcal{G} = (V, E)$, where $V = [n]$ denotes the set of vertices and $E \subseteq V \times V$ the set of edges. We denote by $\mathbf{A}$ its adjacency matrix, meaning that for all $(i, j) \in V^2$, $[\mathbf{A}]_{ij} = 1$ iff $(i, j) \in E$, and by $\mathbf{D}$ the diagonal matrix of vertex degrees. The graph Laplacian of $\mathcal{G}$ is defined as $\mathbf{L} = \mathbf{D} - \mathbf{A}$.

**Setup.** We consider a decentralized setting where $n \geq 2$ real-valued observations $X_1, \ldots, X_n$ are distributed over a communication network represented by a *connected* and *non-bipartite* graph $\mathcal{G} = (V, E)$, see [12]: the observation $X_k$ is assigned to node $k \in [n]$. For simplicity, we assume no ties: for all $k \neq l$, $X_k \neq X_l$. Communication between nodes occurs in a stochastic and pairwise manner: at each iteration, an edge of the communication graph $\mathcal{G}$ is chosen uniformly at random, allowing the corresponding neighboring nodes to exchange information. This popular setup is robust to network changes and helps reduce communication overhead and network congestion [5, 10, 13]. We focus on the *synchronous gossip* setting, where nodes have access to a global clock and synchronize their updates [5, 10, 13]. We assume that a fraction $0 < \varepsilon < 1/2$ of the data is corrupted and may contain outliers, as stipulated in Huber's contamination model, refer to [19].

**The Decentralized Estimation Problem.** The goal pursued here is to develop a robust gossip algorithm that accurately estimates the mean over the network despite the presence of outliers, specifically the $\alpha$-trimmed mean with $\alpha \in (0, 1/2)$, *i.e.,* the average of the middle $(1 - 2\alpha)$-th fraction of the observations. This statistic, discarding the observations of greater or smaller rank, is a widely used location estimator when data contamination is suspected; see [31, 34]. Formally, let $X_{n(1)} \leq X_{n(2)} \leq \cdots \leq X_{n(n)}$ be the order statistics (*i.e.,* the observations sorted in ascending order), and define $m = \lfloor \alpha n \rfloor$ for $\alpha \in (0, 1/2)$. The $\alpha$-trimmed mean is given by:

$$\bar{x}_\alpha = \frac{1}{n - 2m} \sum_{k=m+1}^{n-m} X_{n(k)}. \tag{1}$$

Based on the rank of each node's observation $X_k$, namely $r_k = 1 + \sum_{l=1}^n \mathbb{I}_{\{X_k > X_l\}}$ in the absence of ties, for $k \in [n]$, it can also be formulated as a weighted average of the observations, just like many other robust statistics:

$$\bar{x}_\alpha = \frac{1}{n} \sum_{k=1}^n w_{n,\alpha}(r_k) X_k, \tag{2}$$

where $w_{n,\alpha}$ is a weight function defined as $w_{n,\alpha}(r_k) = (n/(n-2m)) \mathbb{I}_{\{r_k \in I_{n,\alpha}\}}$, with the inclusion interval given by $I_{n,\alpha} = [u, v]$ where $u = m + 1$ and $v = n - m$.

**Remark 1.** *The framework can be extended to the case of $\ell \geq 2$ tied observations with the mid-rank method: the rank assigned to the $\ell$ tied values is the average of the ranks they would have obtained in absence of ties, that is, the average of $p + 1, p + 2, \ldots, p + \ell$, which equals $p + (\ell + 1)/2$, see [24]. To account for the possibility of non-integer rank estimates, one may use the adjusted inclusion interval to define the weights in (2): $I_{n,\alpha} = [u - 1/2, v + 1/2]$. Note that when ranks are integers, this adjustment has no effect.*

**Remark 2.** *Our setup assumes honest nodes, meaning they perform updates correctly and consistently based on their local observations. However, under Huber's contamination model, these observations may include outliers. Note that this setup is fundamentally different from the Byzantine model where nodes may behave arbitrarily or maliciously, potentially sending incorrect updates with the intent to disrupt consensus or degrade performance. This model is still realistic in many practical settings: a sensor could be miscalibrated, stuck at a fixed value, or may consistently report incorrect readings due to environmental factors, without necessarily being attacked.*

## 2.2 Related Works – State of the Art

The overview of related literature below highlights the novelty of the gossip problem analyzed here.

**Distributed Ranking.** Distributed ranking (or ordering) is considered in [9], where a gossip algorithm for estimating ranks on any communication graph is proposed. The algorithm is proved to converge, in the sense of yielding the correct rank estimate, in finite time with probability one. However, their work does not provide any convergence rate bound, and empirical results suggest that the algorithm is suboptimal in scenarios with long return times, *i.e.,* when the expected time for a random walk on the graph to return to its starting node is long (see the Supplementary Material). Alternatively, here we take advantage of the fact that ranks can be calculated using pairwise comparisons $\mathbb{I}_{\{X_k > X_l\}}$, so as to build on the *GoSta* approach in [10], originally introduced for the distributed estimation of $U$-statistics, with a proved convergence rate bound of order $\mathcal{O}(1/t)$.

**Robust Mean Estimation.** To the best of our knowledge, the estimation of $\alpha$-trimmed means has not yet been explored in the gossip literature. Several related works examine the estimation of medians and quantiles in sensor networks [15, 23, 33]. A key limitation of these works is their reliance on a special node, such as a base station or leader, which initiates queries, broadcasts information, and collects data from other sensors—an assumption that does not apply to our setting [15, 23, 33]. Another work has proposed an algorithm for estimating quantiles [16]; however, there seems to be no guarantees of convergence to the true quantiles. Moreover, their algorithm assumes a fully connected communication graph and the ability to sample from four nodes at each step and to sample $K$ random nodes at the end of the protocol, whereas we consider the more challenging setting of arbitrary communication graphs and pairwise communication. Recently, He et al. [17] proposed a novel local aggregator, *ClippedGossip*, for Byzantine-robust gossip learning. In our pairwise setup with a fixed clipping radius, their approach—though reasonable for robust optimization—does not work for robust estimation. Specifically, *ClippedGossip* ultimately converges to the corrupted mean and, therefore, fails to reduce the impact of outliers. A detailed analysis is provided in the Supplementary Material.

The following table assesses whether each method is fully decentralized, whether the estimator is unbiased, and whether theoretical convergence rates exist (for both complete and arbitrary graphs).

| Method | Decentralized? | Unbiased? | Rates on: Any Graph? Complete Graph? |
|---|:---:|:---:|:---:|
| Chiuso et al. (Baseline) | × | ✓ | Complete Graph: $\mathcal{O}(\exp(-t/|E|))$ |
| Baseline++ (ours) | × | ✓ | Complete Graph: $\mathcal{O}(\exp(-t/|E|))$ |
| **GoRank (ours)** | ✓ | ✓ | Any Graph: $\mathcal{O}(1/t)$ |
| Haeupler et al. | × | ∼ | Complete Graph: $\mathcal{O}(\log(n))$ rounds |
| He et al. | ✓ | × | × |
| Shrivastava et al. | × | ✓ | × |
| **GoTrim (ours)** | ✓ | ✓ | Any Graph: $\mathcal{O}(1/t)$ |

## 3 A Gossip Algorithm for Distributed Ranking – GORANK

In this section, we introduce and analyze the GORANK algorithm. We establish that the expected estimates converge to the true ranks at a $\mathcal{O}(1/ct)$ rate, where the constant $c > 0$ (given in Theorem 1 below) quantifies the degree of connectivity of $\mathcal{G}$: the more connected the graph, the greater this quantity and the smaller the rate bound. We also prove that the expected absolute error decreases at a rate of $\mathcal{O}(1/\sqrt{ct})$. In addition, we empirically validate these results with experiments involving graphs of different types, showing that the observed convergence aligns with the theoretical bounds.

### 3.1 Algorithm – Convergence Analysis

We introduce GORANK, a gossip algorithm for estimating the ranks of the observations distributed on the network, see Algorithm 1. It builds on GOSTA, an algorithm originally designed for estimating pairwise averages ($U$-statistics of degree 2) proposed and analyzed in [10]. The GORANK algorithm exploits the fact that ranks can be computed by means of pairwise comparisons:

$$r_k = 1 + n \left( \frac{1}{n} \sum_{l=1}^{n} \mathbb{I}_{\{X_k > X_l\}} \right) = 1 + n r'_k, \text{ for } k = 1 \ldots, n. \tag{3}$$

---

**Algorithm 1** GoRank: a synchronous gossip algorithm for ranking.

---
1: **Init:** For each $k \in [n]$, initialize $Y_k \leftarrow X_k$ and $R'_k \leftarrow 0$.   // `init(k)`
2: **for** $s = 1, 2, \ldots$ **do**
3:   **for** $k = 1, \ldots, n$ **do**
4:     Update estimate: $R'_k \leftarrow (1 - 1/s)R'_k + (1/s)\mathbb{I}_{\{X_k > Y_k\}}$.
5:     Update rank estimate: $R_k \leftarrow nR'_k + 1$.   // `update(k, s)`
6:   **end for**
7:   Draw $(i, j) \in E$ uniformly at random.
8:   Swap auxiliary observation: $Y_i \leftrightarrow Y_j$.   // `swap(i, j)`
9: **end for**
10: **Output:** Estimate of ranks $R_k$.

---

Let $R_k(t)$ and $R'_k(t)$ denote the local estimates of $r_k$ and $r'_k$ respectively at node $k \in [n]$ and iteration $t \geq 1$. Each node maintains an auxiliary observation, denoted $Y_k(t)$, which enables the propagation of observations across the network, despite communication constraints. For each node $k$, the variables are initialized as $Y_k(0) = X_k$ and $R'_k(0) = 0$. At each iteration $t \geq 1$, node $k$ updates its estimate $R'_k(t)$ by computing the running average of $R'_k(t-1)$ and $\mathbb{I}_{\{X_k > Y_k(t-1)\}}$. The rank estimate is then computed as $R_k(t) = nR'_k(t) + 1$. Next, an edge $(i, j) \in E$ is selected uniformly at random, and the corresponding nodes exchange their auxiliary observations: $Y_i(t) = Y_j(t-1)$ and $Y_j(t) = Y_i(t-1)$. This random swapping procedure allows each observation to perform a random walk on the graph, which is described by the permutation matrix $\mathbf{W}_1(t) = \mathbf{I}_n - (\mathbf{e}_i - \mathbf{e}_j)(\mathbf{e}_i - \mathbf{e}_j)^\top$, which plays a key role in the convergence analysis [10]. By taking the expectation with respect to the edge sampling process, we obtain $\mathbf{W}_1 := \mathbb{E}[\mathbf{W}_1(t)] = \mathbf{I}_n - (1/|E|)\mathbf{L}$. This stochastic matrix $\mathbf{W}_1$ shares similarities with the transition matrix used in gossip averaging [5, 25, 32]: it is symmetric and doubly stochastic. Consequently, $\mathbf{W}_1$ has eigenvector $\mathbf{1}_n$ with eigenvalue 1, resulting in a random walk with uniform stationary distribution. In other words, after sufficient iterations (*i.e.,* in a nearly stationary regime), each observation has (approximately) an equal probability of being located at any given node—a property that would naturally hold (exactly) without swapping if the communication graph were complete. In addition, it can be shown that, if the graph is connected and non-bipartite, the spectral gap of $\mathbf{W}_1$ satisfies $0 < c < 1$ and is given by $c = \lambda_2/|E|$ where $\lambda_2$ is the second smallest eigenvalue (or spectral gap) of the Laplacian [10]. This spectral gap plays a crucial role in the mixing time of the random walk, reflecting how quickly it (geometrically) converges to its stationary distribution [32]. In fact, the spectral gap of the Laplacian is also known as the graph's algebraic connectivity, a larger spectral gap meaning a higher graph connectivity [27].

**Remark 3.** *We assume that the network size $n \geq 2$ is known to the nodes. If not, it can be easily estimated by injecting a value $+1$ into the network, where all initial estimates are set to $0$, and applying the standard gossip algorithm for averaging [5], see the Supplementary Material. The quantities computed for each node will then converge to $1/n$.*

**Remark 4.** *The asynchronous extension of* GORANK *is straightforward: it replaces the global iteration counter $s$ with a local counter $C_k$ maintained at each node $k$. The update rule then becomes $(1 - 1/C_k)R'_k + (1/C_k)\mathbb{I}_{X_k > Y_k}$. Empirically, we find that Asynchronous* GORANK *converges slightly faster and is more efficient than its synchronous counterpart. Due to space constraints, the detailed algorithm and experimental results are deferred to Appendix H.*

We now establish convergence results for GORANK, by adapting the analysis in [10], originally proposed to derive convergence rate bounds for GOSTA as follows. For $k \in [n]$, set $\mathbf{h}_k = (\mathbb{I}_{\{X_k > X_1\}}, \ldots, \mathbb{I}_{\{X_k > X_n\}})^\top$ and observe that the true rank of observation $X_k$ is given by $r_k = \mathbf{h}_k^\top \mathbf{1}_n + 1$. At iteration $t = 1$, the auxiliary observation has not yet been swapped, so the expected estimate is updated as $\mathbb{E}[R'_k(1)] = \mathbf{h}_k^\top \mathbf{e}_k$. At the end of the iteration, the auxiliary observation is randomly swapped, yielding the update: $\mathbb{E}[R'_k(2)] = (1/2)\mathbb{E}[R'_k(1)] + (1/2)\mathbf{h}_k^\top \mathbf{W}_1 \mathbf{e}_k$, where $\mathbb{E}[\cdot]$ denotes the expectation taken over the edge sampling process. Using recursion, the evolution of the estimates, for any $t \geq 1$ and $k \in [n]$, is given by

$$\mathbb{E}[R'_k(t)] = (1/t)\sum_{s=0}^{t-1} \mathbf{h}_k^\top \mathbf{W}_1^s \mathbf{e}_k \text{ and } \mathbb{E}[R_k(t)] = n\mathbb{E}[R'_k(t)] + 1.$$

Note that $\mathbb{E}[R'_k(t)]$ can be viewed as the average of $t$ terms of the form $\mathbb{I}_{\{X_k > X_l\}}$, where $X_l$ is picked at random. Observe also that $\mathbf{R}(t) = (R_1(t), \ldots, R_n(t))$ is not a permutation of $[n]$ in general: in particular, we initially have $R_k(1) = 1$ for all $k \in [n]$. We now state a convergence result for GORANK, which claims that the expected estimates converge to the true ranks at a rate of $\mathcal{O}(1/ct)$.

**Theorem 1** (Convergence of Expected GORANK Estimates). *We have: $\forall k \in [n]$, $\forall t \geq 1$,*

$$|\mathbb{E}[R_k(t)] - r_k| \leq \frac{\sigma_k}{ct},$$

*where the constant $c = \lambda_2/|E|$ represents the connectivity of the graph, with $\lambda_2$ being the spectral gap of the graph Laplacian, and the rank functional $\sigma_k = n^{3/2} \cdot \phi((r_k - 1)/n)$ is determined by the score generating function $\phi : u \in (0, 1) \to \sqrt{u(1 - u)}$.*

More details, as well as the technical proof, are deferred to section C of the Supplementary Material. Theorem 1 establishes that, for each node, the estimates converge in expectation to the true ranks at $\mathcal{O}(1/ct)$ rate, where $c$ is a constant depending on the graph's algebraic connectivity: higher network connectivity leads to faster convergence. In addition, the shape of the function $\phi$ involved in the bound (1), resp. of $u \in (0, 1) \mapsto \sqrt{u(1 - u)}$, suggests that extreme (*i.e.*, the lowest and largest) values are intrinsically easier to rank than middle values, see also Fig. 2. Regarding its shape, observe that GORANK can be seen as an algorithm that estimates Bernoulli parameters $(1/n) \sum_{l=1}^{n} \mathbb{I}_{\{X_k > X_l\}} = (r_k - 1)/n$ and, from this perspective, the $\sigma_k$'s can be viewed as the related standard deviations, up to the factor $n^{3/2}$.

We state an additional result below that builds upon the previous analysis, and establishes a bound of order $\mathcal{O}(1/\sqrt{ct})$ for the expected absolute deviation.

**Theorem 2** (Expected Gap). *Let $k \in [n]$, and let $c$ and $\sigma_k$ be as defined in Theorem 1. For all $t \geq 1$, we have: $\mathbb{E}[|R_k(t) - r_k|^2] \leq \mathcal{O}(1/ct) \cdot \sigma_k^2$. Consequently,*

$$\mathbb{E}[|R_k(t) - r_k|] \leq \mathcal{O}\left(\frac{1}{\sqrt{ct}}\right) \cdot \sigma_k \ .$$

Refer to section C in the Supplementary Material for the technical proof. Theorem 2 shows that the expected error in absolute deviation decreases at a rate of $\mathcal{O}(1/\sqrt{ct})$, similar to the convergence rate for the expectation $\mathbb{E}[\mathbf{R}(t)]$, but with a square root dependence. In addition, Theorem 2 can be combined with Markov's inequality to derive high-probability bounds on the ranking error. As will be shown in the next section, this is a key component in the convergence analysis of GOTRIM.

Building upon the previous analysis, we derive another gossip algorithm called *Baseline++*, an improved variant of the one proposed by Chiuso et al. for decentralized rank estimation [9]. It is described at length in section A of the Supplementary Material. The algorithm's steps and propagation closely resemble those of GORANK. However, instead of estimating the means (3) based on pairwise comparisons directly, *Baseline++* aims to minimize, for each node $k \in [n]$, a specific ranking loss function, namely the Kendall $\tau$ distance $\phi_k(\mathbf{X}, \mathbf{R}) = \sum_{l=1}^{n} \mathbb{I}_{\{(X_k - X_l) \cdot (R_k - R_l) < 0\}}$, counting the number of discordant pairs among $((R_k, X_k), (R_l, X_l))$ with $l = 1, \ldots, n$. While GORANK provides a quick approximation of the ranks, especially in poorly connected graphs, *Baseline++*, despite being slower at first, may ultimately achieve a lower overall error as it minimizes a discrete loss function. However, GORANK has a practical advantage: it does not require an initial ranking, which may not always be available in real-world settings. Moreover, it comes with convergence rate guarantees that apply to any graph topology. In contrast, analyzing the convergence of *Baseline++* on arbitrary communication graphs is challenging and is left for future work.

### 3.2 Numerical Experiments

**Setup.** We conduct experiments on a dataset $S = \{1, \ldots, n\}$ with $n = 500$, distributed across nodes of a communication graph. Our evaluation metric is the normalized absolute error between estimated and true ranks, *i.e.*, for node $k$ at iteration $t$, the error is defined as $\ell_k(t) = |R_k(t) - r_k|/n$. While more sophisticated ranking metrics such as Kendall $\tau$ exist, they are not relevant in our context, as we focus on individual rank estimation accuracy. We first examine the impact of the constants appearing in our theoretical bounds (see Theorem 1), the one related to graph connectivity and the one reflecting rank centrality. To this end, we consider three graph topologies: the complete

graph ($c = 4.01 \times 10^{-3}$), in which every node is directly connected to all others, yielding maximal connectivity; the two-dimensional grid ($c = 1.65 \times 10^{-5}$), where each node connects to its four immediate neighbors; and the Watts–Strogatz network ($c = 3.31 \times 10^{-4}$), a randomized graph with average degree $k = 4$ and rewiring probability $p = 0.2$, offering intermediate connectivity between the complete and grid graphs. Then, we compare GORANK with the two other ranking algorithms *Baseline* and *Baseline++*. Figure (a) was generated on a Watts-Strogatz graph, averaged over $1e3$ trials. Figure (b) and (c) show the convergence of ranking algorithms with mean and standard deviation computed from 100 trials: figure (b) shows the convergence of GORANK on different graph topologies; figure (c) compares the convergence of the ranking algorithms on a 2D Grid graph. All experiments were run on a single CPU with 32 GB of memory for $8e4$ iterations, with a total execution time of approximately two hours. The code for our experiments is publicly available.[1]

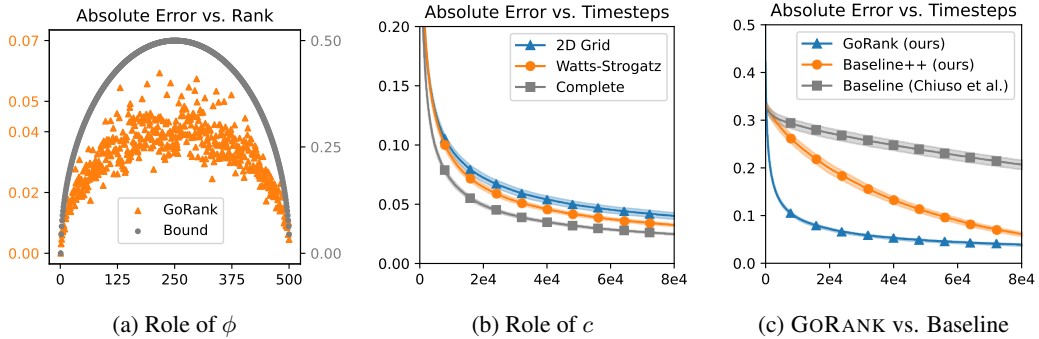

(a) Role of $\phi$          (b) Role of $c$          (c) GORANK vs. Baseline

Figure 2: Illustration of the behavior of GORANK: (a) shows how the absolute error of the rank estimates of GORANK aligns with the shape of the function $\phi$, and highlights the role of the constant $\sigma_k$ for $k \in [n]$ in the error bound; (b) compares the convergence rate of GORANK across different graph topologies (*i.e.,* levels of connectivity), illustrating the influence of the constant $c$ in the bound; (c) compares GORANK with existing method (*Baseline*) and an alternative ranking algorithm developed in this work (*Baseline++*).

**Results.** Together, the figures illustrate key properties of GORANK. Figure (a) empirically confirms that extreme ranks are easier to estimate than those in the middle—consistent with the shape of the theoretical bound. Figure (b) highlights the impact of graph topology: as connectivity decreases, convergence slows, in line with the theoretical bounds. Finally, Figure (c) compares the different methods. GORANK and *Baseline++* provide fast rank approximations even for large, poorly connected graphs (*e.g.,* a 2D grid graph). *Baseline++* appears to converge faster in the end, as it directly optimizes a discrete loss. *Baseline*, on the other hand, converges more slowly throughout, which aligns with its dependence on the return time (*i.e.,* the expected time for a random walk to revisit its starting node), a quantity known to be large for poorly connected graphs. More extensive experiments and discussion, confirming the results above, can be found in the Supplementary Material.

## 4 GOTRIM – A Gossip Algorithm for Trimmed Means Estimation

In this section, we present GOTRIM, a gossip algorithm for trimmed means estimation. We establish a convergence in $\mathcal{O}(1/t)$ with constants depending on the network and data distribution.

### 4.1 Algorithm – Convergence Analysis

We introduce GOTRIM, a gossip algorithm to estimate trimmed mean statistics (see Algorithm 2), which dynamically computes a weighted average using current estimated ranks. Notably, GOTRIM can be paired with any ranking algorithm, including GORANK consequently. Let $Z_k(t)$ and $W_k(t)$ denote the local estimates of the statistic and weight at node $k$ and iteration $t$. First, by Equation (2), the $\alpha$-trimmed mean can be computed via standard gossip averaging: for all $k \in [n]$, $Z_k(t) = (1/n) \sum_{l=1}^{n} W_l(t) \cdot X_l$, where $W_l(t) = w_{n,\alpha}(R_l(t))$ with $w_{n,\alpha}(\cdot)$ defined in Section 2. Secondly,

---

[1]The code is available at github.com/anna-vanelst/robust-gossip.

---

**Algorithm 2** GoTrim: a synchronous gossip algorithm for estimating $\alpha$-trimmed means.

---

1: **Input:** Trimming level $\alpha \in (0, 1/2)$, function $w_{n,\alpha}$ defined in 2 and choice of ranking algorithm rank (*e.g.,* GoRank).
2: **Init:** For all node $k$, set $Z_k \leftarrow 0$, $W_k \leftarrow 0$ and $R_k \leftarrow$ rank.init$(k)$.
3: **for** $s = 1, 2, \ldots$ **do**
4:     **for** $k = 1, \ldots, n$ **do**
5:         Update rank: $R_k \leftarrow$ rank.update$(k, s)$.
6:         Set $W'_k \leftarrow w_{n,\alpha}(R_k)$.
7:         Set $Z_k \leftarrow Z_k + (W'_k - W_k) \cdot X_k$.
8:         Set $W_k \leftarrow W'_k$.
9:     **end for**
10:    Draw $(i, j) \in E$ uniformly at random.
11:    Set $Z_i, Z_j \leftarrow (Z_i + Z_j)/2$.
12:    Swap auxiliary variables: swap(i, j)
13: **end for**
14: **Output:** Estimate of trimmed mean $Z_k$.

---

since ranks $R_k(t)$ vary over iterations, the algorithm dynamically adjusts to correct past errors: at each step, it compensates by injecting $(W_k(t) - W_k(t-1)) \cdot X_k$ into the averaging process. On the one hand, we have an averaging operation which is captured by the averaging matrix: $\mathbf{W}_2(t) = \mathbf{I}_n - (e_i - e_j)(e_i - e_j)^\top/2$. Similarly to the permutation matrix (see the previous section), the expectation of the averaging matrix is symmetric, doubly stochastic and has spectral gap that satisfies $0 < c_2 < 1$ and is given by $c_2 = c/2$ [5, 32]. On the other hand, we have a non-linear operation that depends on the estimated ranks: at each iteration $t > 0$, each node $k$ is updated as $Z_k(t) = Z_k(t-1) + \delta_k(t) \cdot X_k$, where $\delta_k(t) = W_k(t) - W_k(t-1)$. Hence, the evolution of the estimates can be expressed as $\mathbf{Z}(t) = \mathbf{W}_2(t) (\mathbf{Z}(t-1) + \boldsymbol{\delta}(t) \odot \boldsymbol{X})$, where $\mathbf{Z}(t) = (Z_1(t), \ldots, Z_n(t))$ and $\boldsymbol{\delta}(t) = (\delta_1(t), \ldots, \delta_n(t))$. Taking the expectation over the sampling process, the expected estimates are given by: $\mathbb{E}[\mathbf{Z}(t)] = \mathbf{W}_2 (\mathbb{E}[\mathbf{Z}(t-1)] + \Delta \boldsymbol{w}(t) \odot \boldsymbol{X})$ with $\mathbf{W}_2 = \mathbf{I}_n - (1/2|E|)\mathbf{L}$ and $\Delta \boldsymbol{w}(t) = \mathbb{E}[\boldsymbol{\delta}(t)]$. For $t = 1$, since $Z_k(0) = 0$, we have $\mathbb{E}[\boldsymbol{Z}(1)] = \mathbf{W}_2 \Delta \boldsymbol{w}(1) \odot \boldsymbol{X}$. Recursively, for any $t > 0$,

$$\mathbb{E}[\boldsymbol{Z}(t)] = \sum_{s=1}^{t} \mathbf{W}_2^{t+1-s} \Delta \boldsymbol{w}(s) \odot \boldsymbol{X}.$$

We first state a lemma that claims that $W_k(t)$ converges in expectation to $w_{n,\alpha}(r_k)$.

**Lemma 1** (Convergence in Expectation of $W_k(t)$)**.** *Let $\mathbf{R}(t)$ and $\mathbf{W}(t)$ be defined as in Algorithm 1 and Algorithm 2, respectively. For all $k \in [n]$ and $t > 0$, we have:*

$$|\mathbb{E}[W_k(t)] - w_{n,\alpha}(r_k)| \leq \mathcal{O}\left(\frac{1}{\gamma_k^2 ct}\right) \cdot \sigma_k^2,$$

*where $\gamma_k = \min(|r_k - a|, |r_k - b|) \geq 1/2$ with $a = \lfloor \alpha n \rfloor + 1/2$ and $b = n - \lfloor \alpha n \rfloor + 1/2$ being the endpoints of interval $I_{n,\alpha}$. The constants $c$ and $\sigma_k$ are those defined in Theorem 2.*

Further details and the complete proof are provided in Section D of the Supplementary Material. Lemma 1 establishes that, for each node $k$, the estimates of the weight $W_k$ converge in expectation to the true weight $w_{n,\alpha}(r_k)$ at a rate of $\mathcal{O}\left(1/\gamma_k^2 ct\right)$. In addition, this lemma suggests that points closer to the interval endpoints are subject to larger errors, as reflected in the constant $\gamma_k$, which is consistent with the intuition that these points require greater precision in estimating ranks.

Having established the convergence of the weights $W_k(t)$, we now focus on the convergence of the estimates $Z_k(t)$. The following theorem demonstrates the convergence in expectation of GOTRIM when paired with the GORANK ranking algorithm.

**Theorem 3** (Convergence in Expectation of GOTRIM)**.** *Let $\mathbf{Z}(t)$ be defined as in Algorithm 2, and assume the ranking algorithm is Algorithm 1. Then, for any $t > T^* = \min \{t > 1 \,|\, ct > 2 \log(t)\}$,*

$$\|\mathbb{E}[\boldsymbol{Z}(t)] - \bar{x}_\alpha \mathbf{1}_n\| \leq \mathcal{O}\left(\frac{1}{c^2 t}\right) \|\mathbf{K} \odot \mathbf{X}\|,$$

*where $\mathbf{K} = (\sigma_1^2/\gamma_1^2, \ldots, \sigma_n^2/\gamma_n^2)$ and $c, \sigma_k, \gamma_k$ are the constants defined in Lemma 1.*

See Section D of the Supplementary Material for the detailed proof. Theorem 3 shows that, for each node $k$, the estimate of the trimmed mean $Z_k$ converges in expectation to the true trimmed mean $\bar{x}_\alpha$ at a rate of $\mathcal{O}\left(1/c^2 t\right)$. While the presence of the $c^2$ term may appear pessimistic, empirical evidence suggests that the actual convergence may be faster in practice. The vector $\mathbf{K}$ acts as a rank-dependent mask over $\mathbf{X}$, modulating the contribution of each data point $X_k$ to the error bound. Specifically: (1) extreme values are more heavily penalized by the mask, as they come with better rank estimation; (2) values with ranks near the trimming interval endpoints are amplified, as small inaccuracies in rank estimation can lead to disproportionately larger errors.

## 4.2 Robustness Analysis - Breakdown Points

Consider a dataset $S$ of size $n \geq 1$ and $T_n = T_n(S)$ a real-valued statistic based on it. The *breakdown point* for the statistic $T_n(S)$ is defined as $\varepsilon^* = p_\infty/n$, where $p_\infty = \min\{p \in \mathbb{N}^* : \sup_{S'_p} |T_n(S) - T_n(S'_p)| = \infty\}$, the supremum being taken over all corrupted datasets $S'_p$ obtained by replacing $p$ samples in $S$ by arbitrary samples. The breakdown point is a popular notion of robustness [20], corresponding here to the fraction of samples that need to be corrupted to make the statistic $T_n$ arbitrarily large (*i.e.,* "break down"). For example, the breakdown point of the $\alpha$-trimmed mean is given by $\lfloor \alpha n \rfloor/n$, which is approximately $\alpha$. We consider here a generalization of this notion, namely the $\tau$-*breakdown point* by replacing $p_\infty$ with $p_\tau = \min\{p \in \mathbb{N}^* \ , \ \sup_{S'_p} |T_n(S) - T_n(S'_p)| \geq \tau\}$, where $\tau > 0$ is a threshold parameter. Since our algorithm does not compute the exact $\alpha$-trimmed mean, we focus instead on determining the $\tau$-breakdown point $\varepsilon_k^*(t)$ of the *partial $\alpha$-trimmed mean* at iteration $t$ for each node $k$, *i.e.,* the estimate $Z_k(t)$. Given that this quantity was previously shown to converge to the $\alpha$-trimmed mean, we expect that $\varepsilon_k^*(t) \leq \alpha$. The following theorem provides framing bounds for the breakdown point of the estimates of GOTRIM when paired with GORANK.

**Theorem 4.** *Let $\tau > 0$ and $\delta, \alpha \in (0, 1)$. With probability at least $1 - \delta$, the $\tau$-breakdown point $\varepsilon_k^*(t)$ of the partial $\alpha$-trimmed mean at iteration $t > T$ for any node $k$ satisfies*

$$\frac{1}{n} \max\left( \left\lfloor \lfloor \alpha n \rfloor + \frac{1}{2} - \frac{K(\delta)}{\sqrt{t - T}} \right\rfloor, 0 \right) \leq \varepsilon_k^*(t) \leq \frac{\lfloor \alpha n \rfloor}{n} \ ,$$

*where $K(\delta) = \mathcal{O}(1/\delta)$ is a constant and $T = \mathcal{O}(\log(1/\tau\delta))$ represents the time allowed for the mean to propagate.*

See Section E of the Supplementary Material for the technical proof, which relies on the idea that, when estimating a partial $\alpha$-trimmed mean, there are two sources of error: (1) the uncertainty in rank estimation, which can lead to incorrect data points being included in the mean, and (2) the delay from the gossip averaging, which requires a certain propagation time $T$ to update the network estimates. Note that, as $t \to \infty$, Theorem 4 recovers the breakdown point of the exact $\alpha$-trimmed mean.

## 4.3 Numerical Experiments

**Setup.** The experimental setup is identical to that of the previous section, with the key difference being the introduction of corrupted data. Specifically, the dataset $S$ is contaminated by replacing a fraction $\varepsilon = 0.1$ of the values with outliers. We consider two types of corruption, each affecting $\lfloor \varepsilon n \rfloor$ randomly selected data points: (a) scaling, where a value $x$ is changed to $sx$, and (b) shifting, where $x$ becomes $x + s$. While this is a relatively simple form of corruption, it is sufficient to *break down* the classical mean. We measure performance using the absolute error between the estimated and true trimmed mean. For node $k$ at iteration $t$, the error is given by $\ell(t) = (1/n) \sum_k |Z_k(t) - \bar{x}_\alpha|$. Experiments (a) and (b) are run on dataset $S$, corrupted with scaling $s = 10$, using a Watts-Strogatz and a 2D grid graph, respectively. Experiment (c) uses the Basel Luftklima dataset, corrupted with shift $s = 100$. This dataset includes temperature measurements from $n = 105$ sensors across Basel. A graph with connectivity $c = 4.7 \times 10^{-4}$ is constructed by connecting sensors within 1 km of each other. The code and dataset for our experiments is publicly available.[2]

**Results.** Figure (a) empirically confirms that the uncertainty in the weight estimates is highest near the boundaries of the interval, consistent with our theoretical bound. Figures (b) and (c) show that GOTRIM, when combined with GORANK or even alternative ranking methods, quickly approximates

---

[2] The code and Basel Luktklima dataset are available at github.com/anna-vanelst/robust-gossip.

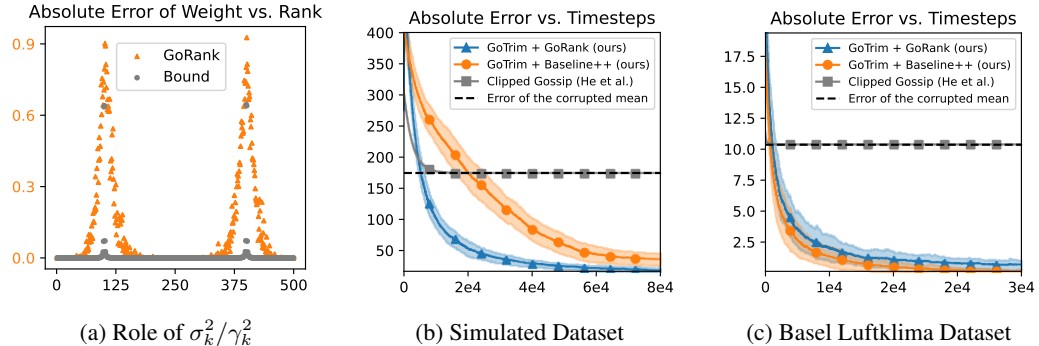

(a) Role of $\sigma_k^2/\gamma_k^2$      (b) Simulated Dataset      (c) Basel Luftklima Dataset

Figure 3: Convergence behavior of GOTRIM in combination with different ranking algorithms. Figure (a) illustrates how the constant in the bound reflects the error of the weight estimate of GOTRIM. Figures (b) and (c) demonstrate that, for $\alpha = 0.2$ and $\varepsilon = 1$, GOTRIM quickly improves on the naive corrupted mean and converges to the trimmed mean.

the trimmed mean and significantly improves over the corrupted mean (indicated by the black dashed line). Overall, GOTRIM quickly outperforms the naive mean under corruption and *ClippedGossip* which will ultimately converge to the corrupted mean. Additional experiments and implementation details are provided in the Supplementary Material.

## 5 Conclusion and Discussion

We introduced and analyzed two novel gossip algorithms: GORANK for rank estimation and GOTRIM for trimmed mean estimation. We proved convergence rates of $\mathcal{O}(1/t)$ and established robustness guarantees for GOTRIM through breakdown point analysis. Empirical results show both methods perform well on large, poorly connected networks: GORANK quickly estimates ranks, and GOTRIM is robust to outliers and improves on the naive mean.

**Byzantine Robustness.** In this work, we focused on robustness to data contamination in the sense of Huber's framework. Extending these results to the more adversarial setting of Byzantine robustness remains an interesting direction. Developing a rigorous theoretical foundation for this setting is still an open problem, and we plan to address it in future work.

**Asynchronous Extension.** Although our analysis focuses on the synchronous setting, real-world systems are often asynchronous. We present the asynchronous version of GORANK in Appendix H. The theoretical analysis in the asynchronous setting is carried out in an extension to this work [35].

**Scalability.** To demonstrate the scalability of our method on large networks, we repeated the experiments from Fig. (c) in Sections 3.2 and 4.3, originally conducted with $n = 500$, on larger networks with $n = 1000$ and $n = 5000$. The detailed results are provided in Appendix I.

**Robustness to Network Disruptions.** While robustness to data contamination is important, the robustness of our proposed algorithms to network disruptions (e.g., edge or node failures, network partitioning) is equally crucial in real-world applications. In Appendix J, we provide a detailed analysis of how our current framework can be extended.

**Performance on Sparse Graphs.** An interesting question is whether our algorithms perform well on sparse graphs. In practice, however, performance depends more on the graph's connectivity than on its sparsity. To illustrate this, we present experiments in Appendix K on sparse graphs with varying levels of connectivity.

**Rank-based Statistics.** GOTRIM naturally extends to the decentralized estimation of rank-based statistics. This includes rank statistics [21], which are key tools in data analysis—particularly for robust hypothesis testing—as well as L-statistics (such as the Winsorized mean).

**Further Discussion.** In appendix L, we provide extended discussion on several topics, including the optimality of the bounds, faster gossip algorithms, extension to multivariate data, and potential applications like robust decentralized optimization.

## Acknowledgments

This research was supported by the PEPR IA Foundry and Hi!Paris ANR Cluster IA France 2030 grants. The authors thank the program for its funding and support.

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

# Outline of the Supplementary Material

The Supplementary Material is organized as follows. Section A introduces two alternative gossip algorithms for distributed ranking: *Baseline* and *Baseline++*. In Section B, we present auxiliary results related to gossip matrices and the convergence analysis of the standard gossip algorithm for averaging. The detailed convergence analysis of GORANK, including the proofs of Theorems 1 and 2, is provided in Section C. Section D presents the convergence proofs for GOTRIM, specifically Lemma 1 and Theorem 3. In Section E, we provide the robustness analysis of GOTRIM (*i.e.,* the proof of Theorem 4). The convergence analysis of *ClippedGossip* is covered in Section F. Section G includes additional experiments and implementation details. Section H introduces Asynchronous GORANK, along with corresponding experimental results. Section I details experiments on larger networks. Section J describes how the current framework can be extended to include network disruptions. Section K provides experiments on sparse networks. Finally, Section L offers further discussion and outlines future work.

## A  Gossip Algorithms for Ranking

Here, we detail two gossip algorithms for ranking, which can be considered natural competitors to GORANK. The algorithm presented first was proposed in [9] and, to our knowledge, is the only decentralized ranking algorithm documented in the literature. The algorithm presented next can be seen as a variant of the latter, incorporating the more efficient communication scheme of GORANK.

### A.1  Baseline: Algorithm from Chiuso et al.

Chiuso et al. propose a gossip algorithm for distributed ranking in a general (connected) network [9]. They demonstrate that this algorithm solves the ranking problem almost surely in finite time. However, they do not provide any convergence rate or non-asymptotic convergence results. The algorithm is outlined in Algorithm 3 and proceeds as follows. At each time step $t$, an edge $(i, j)$ is selected, and the algorithm operates in three phases:

1. **Ranking:** The nodes check if both their local and auxiliary ranks are consistent with the corresponding local and auxiliary observations. If the ranks are inconsistent, the nodes exchange their local and auxiliary rank.

2. **Propagation:** The nodes swap all their auxiliary variables.

3. **Local update:** Each of the two nodes verifies if the auxiliary node has the same ID. If so, it updates its local rank estimate based on the auxiliary rank estimate.

---

**Algorithm 3** Algorithm from Chiuso et al. (Baseline)

---

1: **Require:** Each node with id $I_k = k$ holds observation $X_k$.
2: **Init:** Each node $k$ initializes its ranking estimate $R_k \leftarrow k$ and its auxiliary variables $R_k^v \leftarrow R_k, X_k^v \leftarrow X_k$ and $I_k^v \leftarrow I_k$.
3: **for** $t = 1, 2, \ldots$ **do**
4:     Draw $(i, j)$ uniformly at random from $E$.
5:     **if** $(X_i - X_j) \cdot (R_i - R_j) < 0$ **then**
6:         Swap rankings of nodes $i$ and $j$: $R_i \leftrightarrow R_j$.
7:     **end if**
8:     **if** $(X_i^v - X_j^v) \cdot (R_i^v - R_j^v) < 0$ **then**
9:         Swap rankings of nodes $i$ and $j$: $R_i^v \leftrightarrow R_j^v$.
10:     **end if**
11:     Swap auxiliary variables of nodes $i$ and $j$: $I_i^v \leftrightarrow I_j^v, R_i^v \leftrightarrow R_j^v$ and $X_i^v \leftrightarrow X_j^v$.
12:     **for** $p \in \{i, j\}$ **do**
13:         **if** $I_p^v = I_p$ **then**
14:             Update local ranking estimate: $R_p \leftarrow R_p^v$.
15:         **end if**
16:     **end for**
17: **end for**
18: **Output:** Each node contains the estimate of the ranking.

---

## A.2 Baseline++ - Our Improved Variant Proposal

The algorithm selects an edge $(i, j)$ at each step, and the corresponding nodes check if the auxiliary ranks are consistent with their auxiliary observations. If the ordering is inconsistent, the nodes swap their auxiliary ranks. Then, each node updates its local rank if the ordering of its local observation is inconsistent with the auxiliary observation. In contrast, the algorithm proposed by Chiuso et al. only updates the local estimates when the wandering estimate returns to its originating node. This design can significantly slow down convergence in graphs with low connectivity and long return times.

---

**Algorithm 4** Baseline++

---

1: **Init:** For all $k$, set $R_k \leftarrow k$, $R'_k \leftarrow k$ and $X'_k \leftarrow X_k$.
2: **for** $t = 1, 2, \ldots$ **do**
3:     Draw $(i, j) \in E$ uniformly at random.
4:     **if** $(X'_i - X'_j) \cdot (R'_i - R'_j) < 0$ **then**
5:         Swap rankings: $R'_i \leftrightarrow R'_j$.
6:     **end if**
7:     **for** $p \in \{i, j\}$ **do**
8:         **if** $(X'_p - X_p) \cdot (R'_p - R_p) < 0$ **or** $X'_p = X_p$ **then**
9:             Update local rank: $R_p \leftarrow R'_p$.
10:        **end if**
11:     **end for**
12:     Swap: $R'_i \leftrightarrow R'_j$ and $X'_i \leftrightarrow X'_j$.
13: **end for**
14: **Output:** Estimate of ranks $R_k$.

---

# B   Auxiliary Results

In this section, we present key properties of the transition matrices that will be essential for the proofs of our main theorems. We also present the standard gossip algorithm for mean estimation, along with its convergence results.

## B.1   Properties of Gossip Matrices

**Lemma** (B.1). *Assume the graph* $\mathcal{G} = (V, E)$ *is connected and non-bipartite. Let* $t > 0$. *If at iteration* $t$, *edge* $(i, j)$ *is selected with probability* $p = \frac{1}{|E|}$, *then the transition matrices are given by*

$$\mathbf{W}_1(t) = \mathbf{I}_n - (\mathbf{e}_i - \mathbf{e}_j)(\mathbf{e}_i - \mathbf{e}_j)^\top, \quad \text{(swapping matrix)} \tag{4}$$

$$\mathbf{W}_2(t) = \mathbf{I}_n - \frac{1}{2}(\mathbf{e}_i - \mathbf{e}_j)(\mathbf{e}_i - \mathbf{e}_j)^\top, \quad \text{(averaging matrix)} \tag{5}$$

*For* $\alpha \in \{1, 2\}$, *denote* $\mathbf{W}_\alpha(t) = \mathbf{I}_n - \frac{1}{\alpha}(\mathbf{e}_i - \mathbf{e}_j)(\mathbf{e}_i - \mathbf{e}_j)^\top$. *The following properties hold:*

(a) *The matrices are symmetric and doubly stochastic, meaning that*

$$\mathbf{W}_\alpha(t)\mathbf{1}_n = \mathbf{1}_n, \quad \mathbf{1}_n^\top \mathbf{W}_\alpha(t) = \mathbf{1}_n^\top.$$

(b) *The matrices satisfy the following equalities:*

$$\mathbf{W}_2(t)^2 = \mathbf{W}_2(t), \quad \mathbf{W}_1(t)^2 = \mathbf{I}_n.$$

(c) *For* $\alpha \in \{1, 2\}$, *we have*

$$\mathbf{W}_\alpha = \mathbb{E}[\mathbf{W}_\alpha(t)] = \mathbf{I}_n - \frac{1}{\alpha|E|}\mathbf{L}. \tag{6}$$

(d) *The matrix* $\mathbf{W}_\alpha$ *is also doubly stochastic and it follows that* $\mathbf{1}_n$ *is an eigenvector with eigenvalue 1.*

*(e) The matrix $\tilde{\mathbf{W}}_\alpha \triangleq \mathbf{W}_\alpha - \frac{\mathbf{1}_n \mathbf{1}_n^\top}{n}$ satisfies, by construction, $\tilde{\mathbf{W}}\mathbf{1}_n = 0$, and it can be shown that $\|\tilde{\mathbf{W}}_\alpha\|_{\mathrm{op}} \leq \lambda_2(\alpha)$, where $\lambda_2(\alpha)$ is the second largest eigenvalue of $\mathbf{W}_\alpha$ and $\|\cdot\|_{\mathrm{op}}$ denotes the operator norm of a matrix.*

*(f) The eigenvalue $\lambda_2(\alpha)$ satisfies $0 \leq \lambda_2(\alpha) < 1$ and $\lambda_2(\alpha) = 1 - \frac{\lambda_2}{\alpha|E|}$ where $\lambda_2$ the spectral gap (or second smallest eigenvalue) of the Laplacian.*

*Proof.* The proofs of (a), (b), (d), and (e) are omitted for brevity. For more details, we refer the reader to [5, 10]. The proof of (c) follows from the fact that

$$\mathbb{E}[\mathbf{W}_\alpha(t)] = \frac{1}{|E|} \sum_{(i,j) \in E} \left( \mathbf{I}_n - \frac{1}{\alpha} (\mathbf{e}_i - \mathbf{e}_j)(\mathbf{e}_i - \mathbf{e}_j)^\top \right)$$

and from the definition of the Laplacian matrix $\mathbf{L} = \sum_{(i,j) \in E} (\mathbf{e}_i - \mathbf{e}_j)(\mathbf{e}_i - \mathbf{e}_j)^\top$.
The proof of (f) can be found in [10] (see Lemmas 1, 2 and 3). $\qquad\square$

## B.2 Convergence Analysis of Gossip for Averaging

In the following section, we present classical results on the gossip algorithm for averaging.

---

**Algorithm 5** Gossip algorithm for estimating the standard average [5]

---

1: Each node $k$ initializes its estimate as $Z_k = X_k$.
2: **for** $t = 1, 2, \ldots$ **do**
3:     Randomly select an edge $(i, j)$ from the network.
4:     Update estimates: $Z_i, Z_j \leftarrow \frac{Z_i + Z_j}{2}$.
5: **end for**

---

**Lemma (B.2).** *Let us assume that $\mathcal{G} = (V, E)$ is connected and non bipartite. Then, for $\mathbf{Z}(t)$ defined in Algorithm 5, we have that for all $k \in [n]$ :*

$$\lim_{t \to +\infty} \mathbb{E}[\mathbf{Z}_k(t)] = \bar{X}_n$$

*Moreover, for any $t > 0$,*

$$\left\| \mathbb{E}[\mathbf{Z}(t)] - \bar{X}_n \mathbf{1}_n \right\| \leq e^{-c_2 t} \left\| \mathbf{X} - \bar{X}_n \mathbf{1}_n \right\|$$

*where $c_2 = 1 - \lambda_2(2) > 0$, with $\lambda_2(2) = 1 - \frac{\lambda_2}{2|E|}$.*

*Proof.* The proof is kept brief, as this is a standard result. For more details, we refer the reader to [5, 10]. To prove this lemma, we will primarily use points (e) and (f) from Appendix B.1.

Let $t > 0$. Recursively, we have $\mathbb{E}[\mathbf{Z}(t)] = \mathbf{W}_2^t \mathbf{X}$ where $\mathbf{W}_2 = \mathbf{I}_n - \frac{1}{2|E|}\mathbf{L}$.

Denote $\tilde{\mathbf{W}}_2 \triangleq \mathbf{W}_2 - \frac{\mathbf{1}_n \mathbf{1}_n^\top}{n}$. Noticing that $\frac{\mathbf{1}_n \mathbf{1}_n^\top}{n} \mathbf{X} = \bar{X}_n \mathbf{1}_n$ and that $\tilde{\mathbf{W}}_2^t = \mathbf{W}_2^t - \frac{\mathbf{1}_n \mathbf{1}_n^\top}{n}$, we have

$$\left\| \mathbb{E}[\mathbf{Z}(t)] - \bar{X}_n \mathbf{1}_n \right\| = \left\| \tilde{\mathbf{W}}_2^t \mathbf{X} \right\|.$$

Since $\mathbf{1}_n^\top \tilde{\mathbf{W}}_2 = 0$, it follows that

$$\left\| \mathbb{E}[\mathbf{Z}(t)] - \bar{X}_n \mathbf{1}_n \right\| \leq (\lambda_2(2))^t \left\| \mathbf{X} - \bar{X}_n \mathbf{1}_n \right\|.$$

Setting $c_2 = 1 - \lambda_2(2) > 0$ finishes the proof, as $0 < \lambda_2(2) < 1$. $\qquad\square$

**Lemma (B.3).** *Let us assume that $\mathcal{G} = (V, E)$ is connected and non bipartite. Then, for $\mathbf{Z}(t)$ defined in Algorithm 5, we have that for any $t > 0$ :*

$$\mathbb{P}(\|\mathbf{Z}(t) - \bar{X}_n \mathbf{1}_n\| \geq \varepsilon \|\mathbf{X} - \bar{X}_n \mathbf{1}_n\|) \leq \frac{e^{-c_2 t}}{\varepsilon^2}.$$

*where $c_2 = 1 - \lambda_2(2) > 0$, with $\lambda_2(2) = 1 - \frac{\lambda_2}{2|E|}$.*

*Proof.* The original proof can be found in [5].

Let $t > 0$ and $\varepsilon > 0$. At iteration $t$, we have $\mathbf{Z}(t) = \mathbf{W}_2(t)\mathbf{Z}(t-1)$. Denoting $\mathbf{Y}(t) = \mathbf{Z}(t) - \bar{X}_n \mathbf{1}_n$, it follows that $\mathbf{Y}(t) = \mathbf{W}_2(t)\mathbf{Y}(t-1)$. Taking the conditional expectation, we get

$$\mathbb{E}[\mathbf{Y}(t)^\top \mathbf{Y}(t) \mid \mathbf{Y}(t-1)] = \mathbf{Y}(t-1)^\top \mathbb{E}[\mathbf{W}_2(t)^\top \mathbf{W}_2(t)]\mathbf{Y}(t-1) \leq \lambda_2(2)\|\mathbf{Y}(t-1)\|^2,$$

as $\mathbb{E}[\mathbf{W}_2(t)^\top \mathbf{W}_2(t)] = \mathbb{E}[\mathbf{W}_2(t)] = \mathbf{W}_2$.

By repeatedly conditioning, we obtain the bound

$$\mathbb{E}[\mathbf{Y}(t)^\top \mathbf{Y}(t)] \leq \lambda_2(2)^t \|\mathbf{Y}(0)\|^2.$$

Finally,

$$\mathbb{P}(\|\mathbf{Z}(t) - \bar{X}_n \mathbf{1}_n\| \geq \varepsilon \|\mathbf{Z}(0) - \bar{X}_n \mathbf{1}_n\|) = \mathbb{P}(\|\mathbf{Z}(t) - \bar{X}_n \mathbf{1}_n\|^2 \geq \varepsilon^2 \|\mathbf{Y}(0)\|^2) \tag{7}$$

$$\leq \frac{1}{\varepsilon^2 \|\mathbf{Y}(0)\|^2} \mathbb{E}[\mathbf{Y}(t)^\top \mathbf{Y}(t)] \tag{8}$$

$$\leq \frac{\lambda_2(2)^t}{\varepsilon^2}. \tag{9}$$

**Remark.** *Using the Cauchy-Schwarz inequality $\mathbb{E}[\|\mathbf{Y}\|] \leq \sqrt{\mathbb{E}[\|\mathbf{Y}\|^2]}$, we can also derive*

$$\mathbb{P}(\|\mathbf{Z}(t) - \bar{X}_n \mathbf{1}_n\| \geq \varepsilon \|\mathbf{Z}(0) - \bar{X}_n \mathbf{1}_n\|) \leq \frac{\left(\sqrt{\lambda_2(2)}\right)^t}{\varepsilon}.$$

Setting $c_2 = 1 - \lambda_2(2) > 0$ finishes the proof, as $0 < \lambda_2(2) < 1$. $\qquad\square$

## C  Convergence Analysis of GORANK

In this section, we will prove Theorem 1 and 2.

### C.1  Proof of Theorem 1

**Theorem 1.** *Let $\mathbf{R}(t)$ be defined in GoRank. We have that for all $k \in [n]$ :*

$$\lim_{t \to +\infty} \mathbb{E}[R_k(t)] = r_k. \tag{10}$$

*Moreover, for any $t > 0$,*

$$|\mathbb{E}[R_k(t)] - r_k| \leq \frac{1}{ct} \cdot \sigma_k, \tag{11}$$

*where $c = \frac{\lambda_2}{|E|}$, with $\lambda_2$ being the second smallest eigenvalue of the graph Laplacian (spectral gap), and $\sigma_k = n\tilde{\sigma}_n\left(\frac{r_k-1}{n}\right)$, with $\tilde{\sigma}_n(x) = \sqrt{nx(1-x)}$.*

*Proof.* To prove Theorem 1, we will primarily use points (e) and (f) from Appendix B.1.

Let $k \in [n]$, $t > 0$, and $\bar{h}_k = \frac{1}{n}\mathbf{h}_k^\top \mathbf{1}_n$. From the previous derivation, one has:

$$\mathbb{E}[R_k'(t)] = \frac{1}{t}\sum_{s=0}^{t-1} \mathbf{h}_k^\top \mathbf{W}_1^s \mathbf{e}_k. \tag{12}$$

Denote $\tilde{\mathbf{W}}_1 \triangleq \mathbf{W}_1 - \frac{\mathbf{1}_n \mathbf{1}_n^\top}{n}$. Noticing that $\mathbf{h}_k^\top \frac{\mathbf{1}_n \mathbf{1}_n^\top}{n}\mathbf{e}_k = \bar{h}_k$ and that $\tilde{\mathbf{W}}_1^s = \mathbf{W}_1^s - \frac{\mathbf{1}_n \mathbf{1}_n^\top}{n}$ for $0 \leq s \leq t-1$, we have

$$|\mathbb{E}[R_k'(t)] - \bar{h}_k| \leq \frac{1}{t}\sum_{s=0}^{t-1} |\mathbf{h}_k^\top \tilde{\mathbf{W}}_1^s \mathbf{e}_k|. \tag{13}$$

Since $\mathbf{1}_n^\top \tilde{\mathbf{W}}_1 = 0$, it follows that

$$|\mathbf{h}_k^\top \tilde{\mathbf{W}}_1^s \mathbf{e}_k| \leq (\lambda_2(1))^s \cdot \left\|\mathbf{h}_k - \bar{h}_k \mathbf{1}_n\right\|.$$

Thus,

$$|\mathbb{E}[R'_k(t)] - \bar{h}_k| \leq \frac{1}{t} \sum_{s=0}^{t-1} (\lambda_2(1))^s \left\| \mathbf{h}_k - \bar{h}_k \mathbf{1}_n \right\| \leq \frac{1}{t} \cdot \frac{1}{1 - \lambda_2(1)} \sqrt{n \bar{h}_k (1 - \bar{h}_k)}. \tag{14}$$

since $\lambda_2(1) < 1$ and $\left\| \mathbf{h}_k - \bar{h}_k \mathbf{1}_n \right\| = \sqrt{\sum_i (\mathbb{I}_{\{X_k > X_i\}} - \bar{h}_k)^2}$. Using $1 - \lambda_2(1) = \frac{\lambda_2}{|E|}$ where $\lambda_2$ is the second smallest eigenvalue of the graph Laplacian, it follows that

$$|\mathbb{E}[R'_k(t)] - \bar{h}_k| \leq \frac{1}{t} \frac{|E|}{\lambda_2} \tilde{\sigma}_n(\bar{h}_k), \tag{15}$$

where $\tilde{\sigma}_n(x) = \sqrt{nx(1-x)}$. Plugging $\mathbb{E}[R_k(t)] = n\mathbb{E}[R'_k(t)] + 1$ and $r_k = n\bar{h}_k + 1$ finishes the proof:

$$|\mathbb{E}[R_k(t)] - r_k| \leq \frac{1}{t} \cdot \frac{|E|}{\lambda_2} \cdot n\tilde{\sigma}_n\left(\frac{r_k - 1}{n}\right). \tag{16}$$

$\square$

## C.2 Proof of Theorem 2

We now present a convergence result for the expected gap, which is key to proving the convergence of GOTRIM.

**Theorem 2** (Expected Gap). *Let all $k \in [n]$, and let $c$ and $\sigma_k$ be as defined in Theorem 1. Then, for all $t \geq 1$, we have: $\mathbb{E}[|R_k(t) - r_k|^2] \leq (3/ct) \cdot \sigma_k^2$. Consequently,*

$$\mathbb{E}\left[|R_k(t) - r_k|\right] \leq \sqrt{\frac{3}{ct}} \cdot \sigma_k \ .$$

*Proof.* The proof relies on the Cauchy-Schwarz inequality:

$$\mathbb{E}[|R_k(t) - r_k|] \leq \sqrt{\mathbb{E}[(R_k(t) - r_k)^2]}.$$

Thus, to prove Theorem 2, a convergence result for the variance term $\mathbb{E}\left[(R_k(t) - r_k)^2\right]$ or equivalently $\mathbb{E}\left[(R'_k(t) - \bar{h}_k)^2\right]$ is required. This result is formalized in the following lemma.

**Lemma** (C.1). *We have*

$$\mathbb{E}\left[(R_k(t) - r_k)^2\right] \leq \frac{3}{ct} \cdot \sigma_k^2,$$

*where the constants are defined in Theorem 1.*

*Proof.* Let $t > 0$ and let $k \in [n]$. Using the update rule in *GoRank*, the estimated rank $R'_k(t)$ of agent $k$ at iteration $t$ can be expressed as follows:

$$R'_k(t) = \frac{1}{t} \sum_{s=0}^{t-1} \mathbf{h}_k^\top \mathbf{W}_1(s :)^\top \mathbf{e}_k. \tag{17}$$

where for any $1 \leq s \leq t - 1$, $\mathbf{W}(s :) = \mathbf{W}(s) \ldots \mathbf{W}(1)$ and $\mathbf{W}(0 :) = \mathbf{I}_n$.

For any $t \geq 1$, let us define $\tilde{\mathbf{W}}_1(t)$ as $\tilde{\mathbf{W}}_1(t) \triangleq \mathbf{W}_1(t) - \frac{\mathbf{1}_{n \times n}}{n}$. Note that $\mathbf{W}_1(t)$ are a real symmetric matrices and that $\mathbf{1}_n$ is always an eigenvector of such matrices. Therefore, for any $0 \leq s \leq t - 1$, one has: $\tilde{\mathbf{W}}_1(s :) = \mathbf{W}_1(s :) - \frac{\mathbf{1}_{n \times n}}{n}$. Using this decomposition in (17) yields

$$R'_k(t) - r_k = \frac{1}{t} \sum_{s=0}^{t-1} (\mathbf{h}_k - \bar{h}_k \mathbf{1}_n)^\top \tilde{\mathbf{W}}_1(s :)\mathbf{e}_k \ ,$$

since $\bar{h}_k = \mathbf{h}_k^\top \frac{\mathbf{1}_{n \times n}}{n} \mathbf{e}_k$ and $\mathbf{1}_n^\top \tilde{\mathbf{W}}_1(s) = 0$. The squared gap can thus be expressed as follows:

$$\left(R'_k(t) - \bar{h}_k\right)^2 = \frac{1}{t^2} \sum_{s=0}^{t-1} \sum_{u=0}^{t-1} (\mathbf{h}_k - \bar{h}_k \mathbf{1}_n)^\top \tilde{\mathbf{W}}_1(s:)^\top \mathbf{e}_k (\mathbf{h}_k - \bar{h}_k \mathbf{1}_n)^\top \tilde{\mathbf{W}}_1(u:)^\top \mathbf{e}_k \ ,$$

or equivalently,

$$\left(R'_k(t) - \bar{h}_k\right)^2 = \frac{2}{t^2} \sum_{s<u} (\mathbf{h}_k - \bar{h}_k \mathbf{1}_n)^\top \tilde{\mathbf{W}}_1(s{:})^\top \mathbf{e}_k (\mathbf{h}_k - \bar{h}_k \mathbf{1}_n)^\top \tilde{\mathbf{W}}_1(u{:})^\top \mathbf{e}_k$$

$$+ \frac{1}{t^2} \sum_{u=0}^{t-1} (\mathbf{h}_k - \bar{h}_k \mathbf{1}_n)^\top \tilde{\mathbf{W}}_1(u{:})^\top \mathbf{e}_k (\mathbf{h}_k - \bar{h}_k \mathbf{1}_n)^\top \tilde{\mathbf{W}}_1(u{:})^\top \mathbf{e}_k \ .$$

Denoting $\tilde{\mathbf{h}}_k = \mathbf{h}_k - \bar{h}_k \mathbf{1}_n$, define $v(s) := \tilde{\mathbf{h}}_k^\top \tilde{\mathbf{W}}_1(s{:})^\top \mathbf{e}_k$. We first consider the second term: $\frac{1}{t^2} \sum_{u=0}^{t-1} v(u)^2$. Note that

$$v(u)^2 = \tilde{\mathbf{h}}_k^\top \tilde{\mathbf{W}}_1(u{:})^\top \mathbf{e}_k \mathbf{e}_k^\top \tilde{\mathbf{W}}_1(u{:}) \tilde{\mathbf{h}}_k \ .$$

Let $\mathbf{J} = \frac{1}{n} \mathbf{1}_n \mathbf{1}_n^\top$. Since $\tilde{\mathbf{h}}_k^\top \mathbf{J} = 0$, we can simplify:

$$v(u)^2 = \tilde{\mathbf{h}}_k^\top \mathbf{W}_1(u{:})^\top \mathbf{e}_k \mathbf{e}_k^\top \mathbf{W}_1(u{:}) \tilde{\mathbf{h}}_k \leq \tilde{\mathbf{h}}_k^\top \mathbf{W}_1(u{:})^\top \mathbf{W}_1(u{:}) \tilde{\mathbf{h}}_k.$$

Since the product of two identical permutation matrices is equal to the identity matrix, we obtain $\mathbf{W}_1(u{:})^\top \mathbf{W}_1(u{:}) = \mathbf{I}_n$ and conclude that $\mathbb{E}[v(u)^2] \leq \|\tilde{\mathbf{h}}_k\|^2$. Now let's consider the first term: $\frac{2}{t^2} \sum_{s<u} v(s)v(u)$. Note that for $s < u$,

$$v(s)v(u) = \tilde{\mathbf{h}}_k^\top \tilde{\mathbf{W}}_1(s{:})^\top \mathbf{e}_k \mathbf{e}_k^\top \tilde{\mathbf{W}}_1(u{:}s+1) \tilde{\mathbf{W}}_1(s{:}) \tilde{\mathbf{h}}_k \ .$$

Taking expectation conditional on $\mathbf{W}_1(s{:})$, we have

$$\mathbb{E}[v(s)v(u) \mid \mathbf{W}_1(s{:})] = \tilde{\mathbf{h}}_k^\top \tilde{\mathbf{W}}_1(s{:})^\top \mathbf{e}_k \mathbf{e}_k^\top \mathbb{E}\left[\tilde{\mathbf{W}}_1(u{:}s+1)\right] \cdot \tilde{\mathbf{W}}_1(s{:}) \tilde{\mathbf{h}}_k \ .$$

Since $\mathbb{E}[\tilde{\mathbf{W}}_1(u{:}s+1)] = \tilde{\mathbf{W}}_1^{u-s}$, we obtain the following bound:

$$\mathbb{E}[v(s)v(u) \mid \mathbf{W}_1(s{:})] \leq \lambda_2(1)^{u-s} \cdot \tilde{\mathbf{h}}_k^\top \tilde{\mathbf{W}}_1(s{:})^\top \tilde{\mathbf{W}}_1(s{:}) \tilde{\mathbf{h}}_k \ .$$

Using the previous derivation, we get:

$$\mathbb{E}[v(s)v(u)] \leq \lambda_2(1)^{u-s} \cdot \|\tilde{\mathbf{h}}_k\|^2 \ .$$

Combining the two terms, we obtain:

$$\mathbb{E}\left[\left(R'_k(t) - \bar{h}_k\right)^2\right] \leq \left(\frac{1}{t} + \frac{2}{t^2} \sum_{s<u} \lambda_2(1)^{u-s}\right) \|\tilde{\mathbf{h}}_k\|^2 \ .$$

Note that:

$$\sum_{s<u} \lambda_2(1)^{u-s} \leq \sum_{u=0}^{t-1} \sum_{d=1}^{u} \lambda_2(1)^d \leq \frac{t}{1 - \lambda_2(1)} \ .$$

Recalling $c = 1 - \lambda_2(1)$,

$$\mathbb{E}\left[\left(R'_k(t) - \bar{h}_k\right)^2\right] \leq \left(\frac{1}{t} + \frac{2}{ct}\right) \|\tilde{\mathbf{h}}_k\|^2 \leq \frac{3}{ct} \cdot \|\tilde{\mathbf{h}}_k\|^2 \ .$$

Plugging $\|\tilde{\mathbf{h}}_k\|^2 = \sigma_n(\bar{h}_k)^2$ and using $\mathbb{E}[(R'_k(t) - \bar{h}_k)^2] = \frac{1}{n^2} \mathbb{E}[(R_k(t) - r_k)^2]$ finish the proof. $\quad\square$

## D    Convergence Proofs for GOTRIM

In this section, we will prove Lemma 1 and Theorem 3.

### D.1    Proof of Lemma 1

**Lemma 1.** *Let $\mathbf{R}(t)$ and $\mathbf{W}(t)$ be defined as in Algorithm 1 and Algorithm 2, respectively. For all $k \in [n]$ and $t > 0$, we have:*

$$|\mathbb{E}[W_k(t)] - w_{n,\alpha}(r_k)| \leq \frac{3}{ct} \cdot \frac{\sigma_k^2}{\gamma_k^2 (1 - 2\alpha)}, \tag{18}$$

*where $\gamma_k = \min(|r_k - a|, |r_k - b|) \geq \frac{1}{2}$ with $a = \lfloor \alpha n \rfloor + \frac{1}{2}$ and $b = n - \lfloor \alpha n \rfloor + \frac{1}{2}$ being the endpoints of interval $I_{n,\alpha}$. The constants $c$ and $\sigma_n(\cdot)$ are as defined in Theorem 2.*

*Proof of Lemma 1.* Let $k \in [n]$ and $t > 0$. Denoting $p_k(t) = \mathbb{E}[\mathbb{I}_{\{R_k(t) \in I_{n,\alpha}\}}] = \mathbb{P}(R_k(t) \in I_{n,\alpha})$, we have $\mathbb{E}[W_k(t)] = \frac{p_k(t)}{c_{n,\alpha}}$ where $c_{n,\alpha} = 1 - 2mn^{-1}$ with $m = \lfloor \alpha n \rfloor$.

Hence, we need to show, for $\gamma_k > 0$:

$$\left| \mathbb{P}(R_k(t) \in I_{n,\alpha}) - \mathbb{I}_{\{r_k \in I_{n,\alpha}\}} \right| \leq \mathbb{P}(|R_k(t) - r_k| \geq \gamma_k) \leq \frac{3}{ct} \cdot \frac{\sigma_k^2}{\gamma_k^2}, \tag{19}$$

The right-hand side of the inequality follows directly from Lemma C.1 via an application of Markov's inequality:

$$\mathbb{P}[|R_k(t) - r_k| \geq \gamma_k] = \mathbb{P}[|R_k(t) - r_k|^2 \geq \gamma_k^2] \leq \frac{\mathbb{E}[|R_k(t) - r_k|^2]}{\gamma_k^2}.$$

For the left-hand side, we introduce the interval $I_{n,\alpha} = [a, b]$ and define

$$\gamma_k = \begin{cases} \min(b - r_k, r_k - a), & \text{if } a \leq r_k \leq b, \\ a - r_k, & \text{if } r_k < a, \\ r_k - b, & \text{if } r_k > b. \end{cases}$$

Observe that $\gamma_k \geq \frac{1}{2}$ since $r_k$ is always discrete. We now analyze the three different cases.

Case 1: $a \leq r_k \leq b$. Then, $\left| p_k(t) - \mathbb{I}_{\{r_k \in I_{n,\alpha}\}} \right| = |1 - p_k(t)| = \mathbb{P}(R_k(t) \notin I_{n,\alpha})$. Since we have $\mathbb{P}(|R_k(t) - r_k| \leq \gamma_k) \leq \mathbb{P}(R_k(t) \in I_{n,\alpha})$, it follows that the probability can be upper-bounded as $\mathbb{P}(R_k(t) \notin I_{n,\alpha}) \leq \mathbb{P}(|R_k(t) - r_k| > \gamma_k) \leq \mathbb{P}(|R_k(t) - r_k| \geq \gamma_k)$.

Case 2: $r_k < a$. Here, $\left| p_k(t) - \mathbb{I}_{\{r_k \in I_{n,\alpha}\}} \right| = p_k(t) = \mathbb{P}(R_k(t) \in I_{n,\alpha})$. Since it holds that $\mathbb{P}(|R_k(t) - r_k| < \gamma_k) \leq \mathbb{P}(R_k(t) \notin I_{n,\alpha})$, we obtain $\mathbb{P}(R_k(t) \in I_{n,\alpha}) \leq \mathbb{P}(|R_k(t) - r_k| \geq \gamma_k)$.

Case 3: $r_k > b$ This case follows symmetrically from the previous one.

In all cases, we have $\left| \mathbb{P}(R_k(t) \in I_{n,\alpha}) - \mathbb{I}_{\{r_i \in I_{n,\alpha}\}} \right| \leq \mathbb{P}(|R_k(t) - r_k| \geq \gamma_k)$.

Finally, the result follows from $\mathbb{E}[W_k(t)] = \frac{p_k(t)}{c_{n,\alpha}}$ and $c_{n,\alpha} = 1 - 2mn^{-1} \geq 1 - 2\alpha$.

$\square$

### D.2 Proof of Theorem 3

Now, we will prove the convergence in expectation of the estimates of GoTRIM.

**Theorem 3** (Convergence in Expectation of GoTrim)**.** *Let $\mathbf{Z}(t)$ be defined GoTrim, and assuming the ranking algorithm is GoRank, we have that for all $k \in [n]$ :*

$$\lim_{t \to +\infty} \mathbb{E}[Z_k(t)] = \bar{x}_\alpha. \tag{20}$$

*Moreover, for any $t > T^* = \min\{t > 1 \mid ct > 2\log(t)\}$, we have*

$$\|\mathbb{E}[\mathbf{Z}(t)] - \bar{x}_\alpha \mathbf{1}_n\| \leq \left( \frac{5}{ct} + \frac{4}{ct - 2\log(t)} \right) \cdot \frac{3}{c(1 - 2\alpha)} \cdot \|\mathbf{K} \odot \mathbf{X}\|,$$

*where $\mathbf{K} = \left[ \frac{\sigma_k^2}{\gamma_k^2} \right]_{k=1}^n$ and $c$, $\sigma_k$ and $\gamma_k$ are constants defined in Lemma 1.*

*Proof of Theorem 3.* Recall that for $t > 1$, the expected estimates are characterized recursively as:

$$\mathbb{E}[\mathbf{Z}(t)] = \sum_{s=1}^t \mathbf{W}_2^{t+1-s} \Delta \boldsymbol{w}(s) \odot \mathbf{X}. \tag{21}$$

Denote $\tilde{\mathbf{W}}_2 \triangleq \mathbf{W}_2 - \frac{\mathbf{1}_n \mathbf{1}_n^\top}{n}$ and notice that $\tilde{\mathbf{W}}_2^t = \mathbf{W}_2^t - \frac{\mathbf{1}_n \mathbf{1}_n^\top}{n}$. Denoting $\mathbf{S}(s) = \Delta \boldsymbol{w}(s) \odot \mathbf{X}$, we have have

$$\mathbb{E}[\mathbf{Z}(t)] = \sum_{s=1}^t \frac{1}{n} \mathbf{1}_n \mathbf{1}_n^\top \mathbf{S}(s) + \sum_{s=1}^t \tilde{\mathbf{W}}_2^{t+1-s} \mathbf{S}(s). \tag{22}$$

Since $\forall i, p_i(0) = 0$, the first term can be rewritten as:

$$\sum_{s=1}^{t} \frac{1}{n} \mathbf{1}_n \mathbf{1}_n^\top \mathbf{S}(s) = \sum_{s=1}^{t} \left( \frac{1}{n} \sum_{i=1}^{n} S_i(s) \right) \mathbf{1}_n = \left( \frac{1}{n} \sum_{i=1}^{n} w_i(t) X_i \right) \mathbf{1}_n, \tag{23}$$

where $\Delta w_i(s) = w_i(s) - w_i(s-1)$ with $w_i(s) = \mathbb{E}[W_i(s)]$. This leads to the bound:

$$\|\mathbb{E}[\boldsymbol{Z}(t)] - \bar{x}_\alpha \mathbf{1}_n\| \leq \left\| \left( \frac{1}{n} \sum_{i=1}^{n} w_i(t) X_i \right) \mathbf{1}_n - \bar{x}_\alpha \mathbf{1}_n \right\| + \|\mathbf{R}(t)\|. \tag{24}$$

The first term simplifies as:

$$\left| \frac{1}{n} \sum_{i=1}^{n} \left( \mathbb{E}[W_i(t)] - w_{n,\alpha}(r_i) \right) X_i \right| \cdot \sqrt{n} \leq \frac{1}{\sqrt{n}} \sum_{i=1}^{n} |\mathbb{E}[W_i(t)] - w_{n,\alpha}(r_i)| \cdot |X_i| \tag{25}$$

$$\leq \frac{1}{\sqrt{n}} \sum_{i=1}^{n} \frac{C_i}{t} \cdot |X_i| \tag{26}$$

$$\leq \frac{1}{t} \|\mathbf{C} \odot \mathbf{X}\|, \tag{27}$$

where $C_i = \frac{3}{c} \cdot \frac{\sigma_k^2}{\gamma_i^2 (1 - 2\alpha)}$ comes from Lemma 1 and we denote $\mathbf{C} = [C_1, \ldots, C_n]$. To bound $\mathbf{R}(t)$, we decompose it using an intermediate time step $T = t - \log(t)/c_2 > 0$, where $c_2 = 1 - \lambda_2(2) > 0$. Let $T^* = \min \{t > 1 \mid c_2 t > \log(t)\}$. We obtain for all $t > T^*$,

$$\mathbf{R}(t) = \sum_{s=1}^{T} \tilde{\mathbf{W}}_2^{t+1-s} \mathbf{S}(s) + \sum_{s=T+1}^{t} \tilde{\mathbf{W}}_2^{t+1-s} \mathbf{S}(s).$$

Using Lemma 1, we derive for $s > 1$, $|\Delta w_k(s)| \leq \frac{2C_k}{s-1}$. We obtain $\|\mathbf{S}(1)\| \leq \|\mathbf{C} \odot \mathbf{X}\|$ and for $s > 1$,

$$\|\mathbf{S}(s)\| \leq \sqrt{\sum_{k=1}^{n} \frac{4C_k^2 X_k^2}{(s-1)^2}} \leq \frac{2}{s-1} \|\mathbf{C} \odot \mathbf{X}\|.$$

Applying this bound and using $c_2 = 1 - \lambda_2(2)$ yields:

$$\|\mathbf{R}(t)\| \leq \lambda_2(2)^{t-T} \sum_{s=1}^{T} \lambda_2(2)^{T-s} \|\mathbf{S}(s)\| + \sum_{s=T+1}^{t} \lambda_2(2)^{t-s} \|\mathbf{S}(s)\|$$

$$\leq e^{-c_2(t-T)} \frac{2}{1 - \lambda_2(2)} \|\mathbf{C} \odot \mathbf{X}\| + \frac{1}{T} \frac{2}{(1 - \lambda_2(2))} \|\mathbf{C} \odot \mathbf{X}\|$$

$$\leq \left( \frac{2}{c_2 t} + \frac{2}{c_2 t - \log(t)} \right) \|\mathbf{C} \odot \mathbf{X}\|.$$

Finally, we establish the following error bound:

$$\|\mathbb{E}[\boldsymbol{Z}(t)] - \bar{x}_\alpha \mathbf{1}_n\| \leq \frac{1}{t} \|\mathbf{C} \odot \mathbf{X}\| + \frac{4}{ct} \|\mathbf{C} \odot \mathbf{X}\| + \frac{2}{\frac{c}{2} t - \log(t)} \|\mathbf{C} \odot \mathbf{X}\|,$$

where $c = 2c_2$. This bound simplifies to

$$\|\mathbb{E}[\boldsymbol{Z}(t)] - \bar{x}_\alpha \mathbf{1}_n\| \leq \left( \frac{5}{ct} + \frac{4}{ct - 2\log(t)} \right) \|\mathbf{C} \odot \mathbf{X}\|.$$

Moreover, we have $\|\mathbf{C} \odot \mathbf{X}\| = \frac{3}{c(1-2\alpha)} \|\mathbf{K} \odot \mathbf{X}\|$ where $\mathbf{K} = \left[ \frac{\sigma_k^2}{\gamma_k^2} \right]_{k=1}^{n}$, which completes the proof. $\qquad \square$

# E  Robustness Analysis: Breakdown Point of the Partial Trimmed Mean

Here, we prove provide a breakdown point analysis of GOTRIM and prove Theorem 4.

**Theorem 4** (Breakdown Point). *Let $\tau > 0$ and $\delta, \alpha \in (0, 1)$. Denote $m = \lfloor \alpha n \rfloor$. With probability at least $1 - \delta$, the $\tau$-breakdown point $\varepsilon_i^*(t)$ of the partial $\alpha$-trimmed mean at iteration $t > T$ for any node $i$ satisfies*

$$\frac{1}{n} \max\left( \left\lfloor m + \frac{1}{2} - \frac{K(\delta)}{\sqrt{t - T}} \right\rfloor, 0 \right) \leq \varepsilon_i^*(t) \leq \frac{m}{n} \ ,$$

*where $T = \frac{4}{c} \log\left(\frac{n}{\varepsilon \delta}\right)$ denote a propagation time with $c$ being the connectivity constant and $\varepsilon := \tau / \max_i |X_i|$ denote a tolerance parameter. Moreover, we define $K(\delta) := \frac{c_m}{\left(1 - \frac{1}{n}\right) \sqrt{c} \delta}$ where $c_m = \sqrt{3} n^{3/2} \cdot \phi((m-1)/n)$ with $\phi : u \in (0, 1) \to \sqrt{u(1 - u)}$.*

*Proof.* Applying Lemma 1, we estimate the breakdown point while accounting for the uncertainty in rank estimation, and combine it with Lemma 2 via a union bound. Lemma 2 introduces $T$ the number of iterations required for the mean to propagate and the maximum delay $T$ corresponds to the smallest possible $\varepsilon$, given by $\varepsilon = \frac{\tau}{B}$, where $B = \max_i |X_i|$. Setting $\delta_1 = \left(1 - \frac{1}{n}\right) \delta$ and $\delta_2 = \frac{\delta}{n}$ complete the proof.

**Lemma E.1** (Instant Breakdown Point). *Let $\delta_1 > 0$. Denote $\tilde{\varepsilon}^*(t)$ the breakdown point at iteration $t$ as the statistic defined as*

$$\frac{1}{n} \sum_{i=1}^{n} w_{n,\alpha}(R_i(t)) X_i.$$

*Then, with probability at least $1 - \delta_1$,*

$$\tilde{\varepsilon}^*(t) \geq \frac{\max\left( \left\lfloor m + \frac{1}{2} - \frac{c_m}{\delta_1 \sqrt{ct}} \right\rfloor, 0 \right)}{n},$$

*where we define $m = \lfloor n\alpha \rfloor$ and $c_m = \sqrt{3} n^{3/2} \cdot \phi((m-1)/n)$. As $t \to \infty$, note that $p = \lfloor \frac{1}{2} + m \rfloor = m$, which allows us to recover the breakdown point of the $\alpha$-trimmed mean.*

*Proof.* We are interested in the observations $X_k$ with rank $1 \leq r_k \leq m$ (the outliers that should be excluded from the mean), where $m = \lfloor n\alpha \rfloor$. Note that we do not consider the data points with rank $n - m + 1 \leq r_k \leq n$, as this case is symmetrical for determining the breakdown point.

Let $1 \leq r_k \leq m$ and consider the probability $p_k(t)$ of including $X_k$ in the mean. This probability satisfies:

$$p_k(t) \leq \frac{\sqrt{3} \sigma_n(r_k)}{(a - r_k) \sqrt{ct}} \leq \frac{c_m}{(a - r_k) \sqrt{ct}},$$

where $a = m + \frac{1}{2}$ is the left endpoint of the inclusion interval. The breakdown point can be interpreted as $\frac{p}{n}$ where $p$ is the maximum rank required to "break" the mean and $n$ is the sample size. To determine the maximum $r_k$ in $[1, m]$ such that the probability of inclusion is lower than a certain confidence parameter $\delta$, we solve for $r_k$:

$$p_k(t) \leq \frac{c_m}{(a - r_k) \sqrt{ct}} \leq \delta,$$

and obtain

$$p = \max\left( \left\lfloor \frac{1}{2} + m - \frac{c_m}{\delta \sqrt{ct}} \right\rfloor, 0 \right).$$

Thus, with probability at least $1 - \delta$, the breakdown point is given by $\tilde{\varepsilon}^*(t) = \frac{p}{n}$.

**Lemma E.2** (Propagation Time). *Let $\delta_2 > 0$ be a confidence parameter and $\varepsilon > 0$ a tolerance parameter. We define $\mathbf{Z}(\cdot)$ as the evolution of the standard mean estimate, initialized as $\mathbf{Z}(0) = \mathbf{X}$. Additionally, we introduce a perturbed estimate, $\tilde{\mathbf{Z}}(\cdot)$, which starts at $\tilde{\mathbf{Z}}(0) = \mathbf{X} + B\mathbf{e}_j$. At iteration $t$, we detect a mistake of magnitude $B$, we correct it by injecting $-B$, leading to the update*

$\tilde{\mathbf{Z}}(t) \leftarrow \tilde{\mathbf{Z}}(t) - B\mathbf{e}_j$. *Then, it follows that, with probability at least $1 - \delta_2$, for all $s \geq t + T$, for any node $i$,*

$$|Z_i(s)| \leq \|\tilde{\mathbf{Z}}(s) - \mathbf{Z}(s)\| \leq \varepsilon B,$$

*where $T$ represents number of iterations required to correct a mistake with tolerance $\varepsilon > 0$ and is given by $T = \frac{4}{c} \log\left(\frac{1}{\varepsilon\delta_2}\right)$ with $c$ is the connectivity of the graph.*

*Proof.* At iteration $t$, the perturbed estimate is $\tilde{\mathbf{Z}}(t) = \mathbf{W}_2(t:)\tilde{\mathbf{Z}}(0) = \mathbf{Z}(t) + \mathbf{W}_2(t:)B\mathbf{e}_j$ since $\mathbf{Z}(t) = \mathbf{W}_2(t:)\mathbf{Z}(0)$. Then, at iteration $s > t$, we inject $-B$ at node $j$ which gives the following: $\tilde{\mathbf{Z}}(s) = \mathbf{W}_2(s:t)\left[\mathbf{Z}(t) + \mathbf{W}_2(t:)B\mathbf{e}_j - B\mathbf{e}_j\right]$. Thus, for any $s > t$,

$$\boldsymbol{\Delta}(s) := \tilde{\mathbf{Z}}(s) - \mathbf{Z}(s) = \mathbf{W}_2(s:t)\left[\mathbf{W}_2(t:) - \mathbf{I}_n\right]B\mathbf{e}_j = \mathbf{W}_2(s)\boldsymbol{\Delta}(s-1).$$

Following the proof in [5], by repeatedly conditioning, we obtain

$$\mathbb{E}[\boldsymbol{\Delta}(s)^\top \boldsymbol{\Delta}(s) \mid \boldsymbol{\Delta}(t)] \leq \lambda_2(2)^{s-t}\|\boldsymbol{\Delta}(s)\|^2.$$

Since $\|\boldsymbol{\Delta}(s)\| \leq B$ and taking the expectation on both sides, we have

$$\mathbb{E}[\boldsymbol{\Delta}(s)^\top \boldsymbol{\Delta}(s)] \leq \lambda_2(2)^{s-t}B^2.$$

Finally, we derive

$$\mathbb{P}(\|\tilde{\mathbf{Z}}(s) - \mathbf{Z}(s)\| \geq \varepsilon B) = \mathbb{P}(\|\tilde{\mathbf{Z}}(s) - \mathbf{Z}(s)\|^2 \geq \varepsilon^2 B^2) \tag{28}$$

$$\leq \frac{1}{\varepsilon^2 B^2}\mathbb{E}[\boldsymbol{\Delta}(s)^\top \boldsymbol{\Delta}(s)] \tag{29}$$

$$\leq \frac{\lambda_2(2)^{s-t}}{\varepsilon^2}. \tag{30}$$

Let $\delta_2 > 0$. We need to solve $\frac{1}{\varepsilon^2}e^{-c_2 T} \leq \delta$, which gives $T = \frac{4}{c} \log\left(\frac{1}{\varepsilon\delta_2}\right)$. It follows that with probability at least $1 - \delta_2$, for $s \geq t + T$, for all nodes $i$,

$$|Z_i(t)| \leq \|\tilde{\mathbf{Z}}(s) - \mathbf{Z}(s)\| \leq \varepsilon B.$$

# F    Convergence Analysis of *ClippedGossip*

In this section, we show that *ClippedGossip*, in our pairwise setup with a fixed clipping radius, ultimately converges to the corrupted mean. The following lemma describes the update rule of *ClippedGossip* through its transition matrix.

**Lemma (Transition Matrix).** *Assume at iteration $t$, edge $(i, j) \in E$ is selected. Then, the update using ClippedGossip rule with constant clipping radius $\tau$ is given by $\boldsymbol{x}^{t+1} = \boldsymbol{W}(t)\boldsymbol{x}^t$ where $\boldsymbol{W}(t) \in \mathbb{R}^{n \times n}$ denotes the transition matrix at iteration $t$ which is given by*

$$\boldsymbol{W}(t) = \boldsymbol{I}_n - \alpha_{ij}^t \boldsymbol{L}_{ij},$$

*where $\boldsymbol{L}_{ij} := (\boldsymbol{e}_i - \boldsymbol{e}_j)(\boldsymbol{e}_i - \boldsymbol{e}_j)^\top$ is the elementary Laplacian associated with edge $(i, j)$ and $\alpha_{ij}^t := \frac{1}{2} \min\left(1, \tau/\|x_i^t - x_j^t\|\right)$ a constant that depends on the previous estimates. Moreover, we have $\boldsymbol{W}(t)\boldsymbol{W}(t)^\top = \boldsymbol{I}_n - 2\alpha_{ij}^t(1 - \alpha_{ij}^t)\boldsymbol{L}_{ij}$. In the special case $\alpha_{ij} = 1/2$, the matrix reduces to the standard averaging and is a projection matrix: $\boldsymbol{W}(t)\boldsymbol{W}(t)^\top = \boldsymbol{W}(t)$.*

*Proof.* At iteration $t$, if edge $(i, j) \in E$ is selected, the update rule is

$$x_i^{t+1} = x_i^t + \frac{1}{2} \text{CLIP}(x_j^t - x_i^t, \tau), \quad x_j^{t+1} = x_j^t + \frac{1}{2} \text{CLIP}(x_i^t - x_j^t, \tau),$$

where the clipping operator is defined as $\text{CLIP}(z, \tau) := \min\left(1, \tau/\|z\|\right)z$. This update can be expressed in matrix form as $\boldsymbol{x}^{t+1} = \boldsymbol{W}(t)\boldsymbol{x}^t$. It equals the identity matrix except for a $2 \times 2$ block corresponding to nodes $i$ and $j$, given by

$$\begin{bmatrix} 1 - \alpha_{ij}^t & \alpha_{ij}^t \\ \alpha_{ij}^t & 1 - \alpha_{ij}^t \end{bmatrix}, \quad \text{where} \quad \alpha_{ij}^t := \frac{1}{2} \min\left(1, \frac{\tau}{\|x_i^t - x_j^t\|}\right).$$

Observe that $\boldsymbol{L}_{ij}^2 = 2\boldsymbol{L}_{ij}^2$ and conclude.

The next lemma allows us to bound the coefficients $\alpha_{ij}^t$ of the transition matrix.

**Lemma (Lower Bound on Coefficient of the Transition Matrix).** *Let $m^t = \max_{(k,l)\in E} \|x_k^t - x_l^t\|$. We have $m^t \leq m$ and thus we derive*

$$\alpha \leq \min\left(1, \frac{\tau}{\|x_i^t - x_j^t\|}\right) \leq 1,$$

*where $\alpha := \min(1, \tau/m)/2$ with $m = \max_{(k,l)\in E} \|x_k^0 - x_l^0\|$.*

*Proof.* Let $m^1 = \max_{(k,l)\in E} \|x_k^1 - x_l^1\|$. We can show that $m^1 \leq m$. After update, we have $x_i^1 = (1 - \alpha_{ij})x_i^0 + \alpha_{ij}x_j^0$ and $x_j^1 = (1 - \alpha_{ij})x_j^0 + \alpha_{ij}x_i^0$. Observe that the new estimates remain within the convex hull of the original points. Since only $x_i$ and $x_k$ have changed, the next maximum distance is:

$$m^1 = \max\left\{ \max_{\substack{k\neq i,j \\ \ell\neq i,j}} \|x_k^0 - x_\ell^0\|, \max_{k\neq i,j} \|x_i^1 - x_k^0\|, \max_{k\neq i,j} \|x_j^1 - x_k^0\|, \|x_i^1 - x_j^1\| \right\}.$$

Note: $\|x_k^0 - x_\ell^0\| \leq m$ by definition. So we just need to show that the other terms are smaller than $m$. For $k \neq i, j$, recall $x_i^1 = (1 - \alpha_{ij})x_i^0 + \alpha_{ij}x_j^0$, then we have

$$\|x_i^1 - x_k^0\| = \|(1 - \alpha_{ij})(x_i^0 - x_k^0) + \alpha(x_j^0 - x_k^0)\| \leq (1 - \alpha_{ij})m + \alpha_{ij}m = m,$$

by the triangle inequality. Similarly, we derive for $k \neq i,j$, $\|x_j^1 - x_k^0\| \leq m$. Finally, since $0 < 1 - 2\alpha_{ij} < 1$, we have

$$\|x_i^1 - x_j^1\| = |1 - 2\alpha_{ij}| \cdot \|x_i^0 - x_j^0\| \leq m.$$

Recursively, we have for all $t \geq 1$, $m^t \leq m$, which finishes the proof.

The next lemma shows that at each iteration, the expected gap between the estimates and the corrupted mean will decrease.

**Lemma (Contraction Bound).** *Let $t > 0$. Define $\boldsymbol{y}^t = \boldsymbol{x}^t - \bar{x}\mathbf{1}_n$ where $\bar{x} := (1/n)\sum_{i=1}^n x_i^0$ is the initial average. We have the following bound:*

$$\mathbb{E}[\|\boldsymbol{y}^{t+1}\|^2 \mid \boldsymbol{y}^t] \leq \left(1 - \frac{\beta\lambda_2}{|E|}\right)\|\boldsymbol{y}^t\|^2.$$

*where $\beta = 2\alpha(1-\alpha)$ with $\alpha := \min(1, \tau/m)/2$ for $m = \max_{(k,l)\in E} \|x_k^0 - x_l^0\|$ and $\lambda_2$ denotes the spectral gap of the Laplacian.*

*Proof.* Taking the conditional expectation over the sampling process, we have

$$\mathbb{E}[(\boldsymbol{y}^{t+1})^\top \boldsymbol{y}^{t+1} \mid \boldsymbol{y}^t] = (\boldsymbol{y}^t)^\top \mathbb{E}[\boldsymbol{W}(t)^\top \boldsymbol{W}(t) \mid \boldsymbol{y}^t]\boldsymbol{y}^t.$$

Since $\boldsymbol{y}^t \perp \mathbf{1}$, and $\mathbf{1}$ is the eigenvector corresponding to the largest eigenvalue 1 of $\mathbb{E}[\boldsymbol{W}(t)^\top \boldsymbol{W}(t) \mid \boldsymbol{y}^t]$,

$$(\boldsymbol{y}^t)^\top \mathbb{E}[\boldsymbol{W}(t)^\top \boldsymbol{W}(t) \mid \boldsymbol{y}^t]\boldsymbol{y}^t \leq \lambda_2\left(\mathbb{E}[\boldsymbol{W}(t)^\top \boldsymbol{W}(t) \mid \boldsymbol{y}^t]\right)\|\boldsymbol{y}^t\|^2.$$

Now from the first lemma, we have $\boldsymbol{W}(t)^\top \boldsymbol{W}(t) = \boldsymbol{I}_n - \beta_{ij}^t \boldsymbol{L}_{ij}$ where $\beta_{ij}^t = 2\alpha_{ij}^t(1 - \alpha_{ij}^t)$. We see that $\beta \leq \beta_{ij}^t \leq 1/2$ where $\beta = 2\alpha(1-\alpha)$. Observing that $\mathbb{E}[\boldsymbol{W}(t)^\top \boldsymbol{W}(t) \mid \boldsymbol{y}^t] = \boldsymbol{I}_n - (1/|E|)\tilde{\boldsymbol{L}}^t$ where we define the weighted Laplacian as

$$\tilde{\boldsymbol{L}}^t := \sum_{(i,j)\in E} \beta_{ij}^t \boldsymbol{L}_{ij}.$$

Using the quadratic form of the weighted Laplacian:

$$\boldsymbol{x}^\top \boldsymbol{L}(w)\boldsymbol{x} = \sum_{(i,j)\in E} w_{ij}[x(i) - x(j)]^2,$$

we derive $\beta \boldsymbol{L} \preceq \tilde{\boldsymbol{L}}^t \preceq (1/2)\boldsymbol{L}$, as $\beta \leq \beta_{ij}^t \leq 1/2$. Using the Courant-Fischer Theorem, we obtain: $\beta \lambda_2 \leq \rho^t \leq \lambda_2/2$, where $\rho^t$ denotes the second smallest eigenvalue of $\tilde{\boldsymbol{L}}^t$ and $\lambda_2$ is the second smallest eigenvalue of the (unweighted) Laplacian $\boldsymbol{L}$. We therefore obtain a bound on the second largest eigenvalue of $\mathbb{E}[\boldsymbol{W}(t)^\top \boldsymbol{W}(t) \mid \boldsymbol{y}^t]$:

$$1 - \frac{\lambda_2}{2|E|} \leq 1 - \frac{\rho^t}{|E|} \leq 1 - \frac{\beta \lambda_2}{|E|},$$

which finishes the proof.

Finally, the convergence of *ClippedGossip* estimates is established in the following proposition.

**Proposition (Convergence of *ClippedGossip* Estimates).** *For each node $k$,*

$$\lim_{t \to +\infty} x_k(t) = \bar{x}.$$

*Moreover, we have*

$$\mathbb{E}[\|\boldsymbol{x}^t - \bar{x}\boldsymbol{1}_n\|^2] \leq \left(1 - \frac{\beta \lambda_2}{|E|}\right)^t \|\boldsymbol{x}^0 - \bar{x}\boldsymbol{1}_n\|^2,$$

*where $\bar{x} := (1/n)\sum_{i=1}^n x_i^0$ is the initial average, $\beta = 2\alpha(1 - \alpha)$ with $\alpha := \min(1, \tau/m)/2$ and $m = \max_{(k,l) \in E} \|x_k^0 - x_l^0\|$ and $\lambda_2$ denotes the spectral gap of the Laplacian.*

*Proof.* Using the previous lemma, recursively, we obtain

$$\mathbb{E}[\|\boldsymbol{y}^t\|^2] \leq \left(1 - \frac{\beta \lambda_2}{|E|}\right)^t \|\boldsymbol{y}^0\|^2.$$

Recalling $\boldsymbol{y}^t = \boldsymbol{x}^t - \bar{\boldsymbol{x}}\boldsymbol{1}_n$ finishes the proof.

# G   Additional Experiments and Implementation Details

This section provides additional experiments and more details on the Basel Luftklima dataset as well as compute resources.

## G.1   Experiments Compute Resources

The experiments are run on a single CPU with 32 GB of memory.

The execution time for each experiment is less than 30 minutes (except for the large-scale experiments). The details of a few experiments are given in Table 1.

| Experiments | Execution Time | Figure |
|---|---|---|
| exp1+exp2+exp3 | $\sim$ 30 min | Ranking (a) |
| exp4+exp5+exp6 | $\sim$ 5 min | Ranking (b) |
| exp7+exp8+exp9 | $\sim$ 15 min | Ranking (c) |
| exp10+exp10a+exp10b | $\sim$ 50 min | Trimmed Mean (a) |
| exp11+exp12+exp13 | $\sim$ 15 min | Trimmed Mean (b) |
| exp14 | $\sim$ 5 min | Trimmed Mean (c) |
| exp15+exp16+exp17 | $\sim$ 5 min | Ranking (d) |

Table 1: Execution times for all experiments

## G.2   Basel Luftklima Dataset

The dataset contains temperature measurements from 99 Meteoblue sensors across the Basel region, recorded between April 14 and April 15, 2025. For each sensor, only the first observation is used. A graph is built by connecting sensors that are within 1 km of each other, based on their geographic coordinates. Only the connected component of the graph is kept. To avoid ties during ranking, we add a small, imperceptible amount of noise to the data.

### G.3 Additional Experiments

Figures 4, 5, and 6 extend the experiments presented in sections 3 and 4.

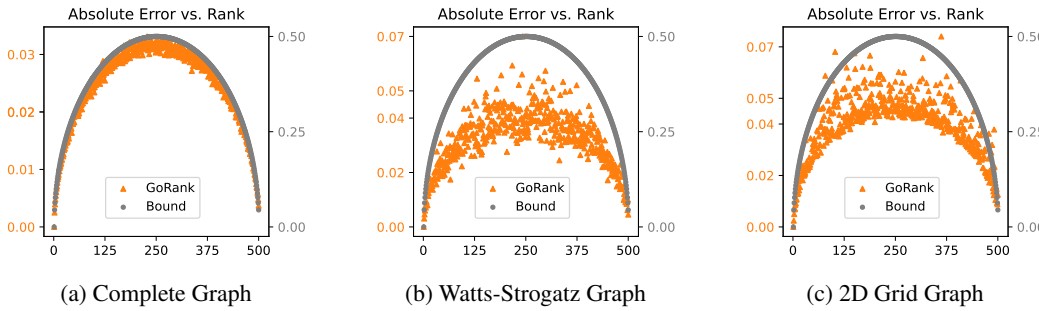

(a) Complete Graph      (b) Watts-Strogatz Graph      (c) 2D Grid Graph

Figure 4: Role of $\phi$ for three different communication graphs

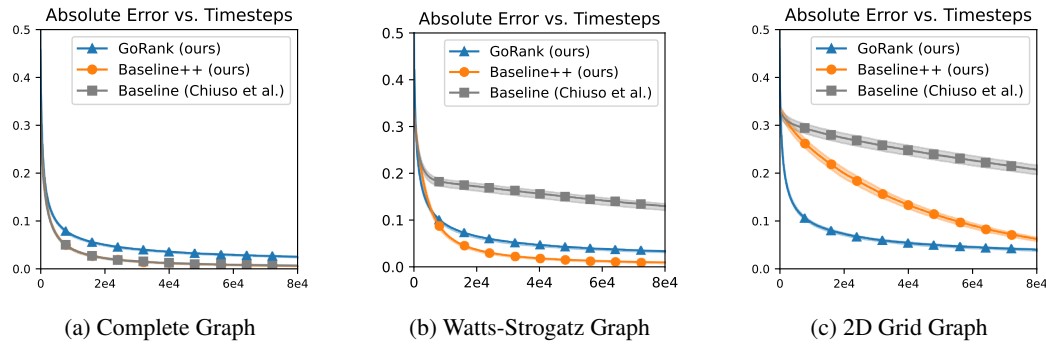

(a) Complete Graph      (b) Watts-Strogatz Graph      (c) 2D Grid Graph

Figure 5: Comparison of ranking algorithms on three different communication graphs

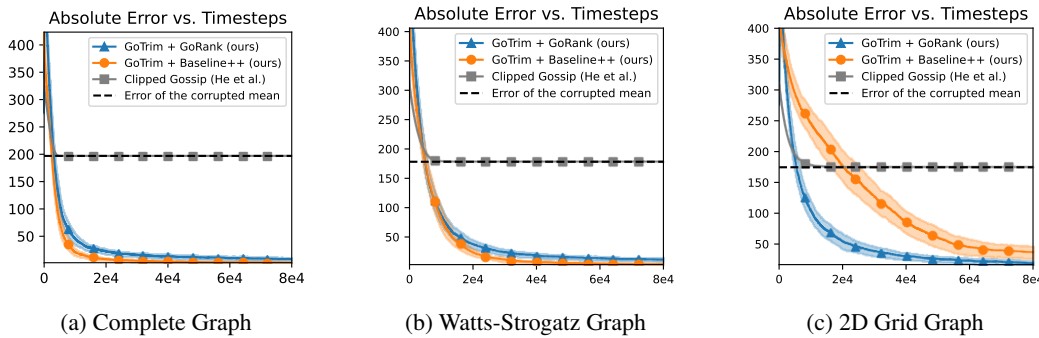

(a) Complete Graph      (b) Watts-Strogatz Graph      (c) 2D Grid Graph

Figure 6: Comparison of gossip algorithms for robust mean estimation on three different graphs

## H  Asynchronous Variant of GORANK

In this section, we propose an asynchronous variant of GORANK that operates without access to a global clock or a shared iteration counter. Instead, each node $k$ maintains a local counter $C_k$ which tracks the number of times it has participated in an update. Asynchronous GORANK, outlined in Algorithm 6 proceeds as follows. When node $k$ is selected, it increments $C_k$ and updates its local rank estimate using a running average, where the global iteration $t$ is replaced by $C_k$. As in the synchronous version of *GoRank*, selected nodes also exchange their auxiliary observations. Note that this asynchronous version is more efficient, as it significantly reduces the number of updates—performing only two updates per iteration instead of $n$.

**Algorithm 6** Asynchronous GoRank

1: **Init:** For each $k \in [n]$, $Y_k \leftarrow X_k$, $R'_k \leftarrow 0$, $C_k \leftarrow 0$.
2: **for** $t = 1, 2, \dots$ **do**
3:   Draw $(i, j) \in E$ uniformly at random.
4:   **for** $p \in \{i, j\}$ **do**
5:     Set $C_p \leftarrow C_p + 1$.
6:     Set $R'_p \leftarrow (1 - 1/C_p)R'_p + (1/C_p)\mathbb{I}_{\{X_p > Y_p\}}$.
7:     Update rank estimate: $R_p \leftarrow nR'_p + 1$.
8:   **end for**
9:   Swap auxiliary observation: $Y_i \leftrightarrow Y_j$.
10: **end for**
11: **Output:** Estimate of ranks $R_k$.

Figure 7 shows a comparison of Asynchronous GoRank with our two other ranking algorithms: Synchronous GoRank and Baseline++. The results suggest that Asynchronous GoRank converges slightly faster than Synchronous GoRank, thereby outperforming it in both speed and efficiency.

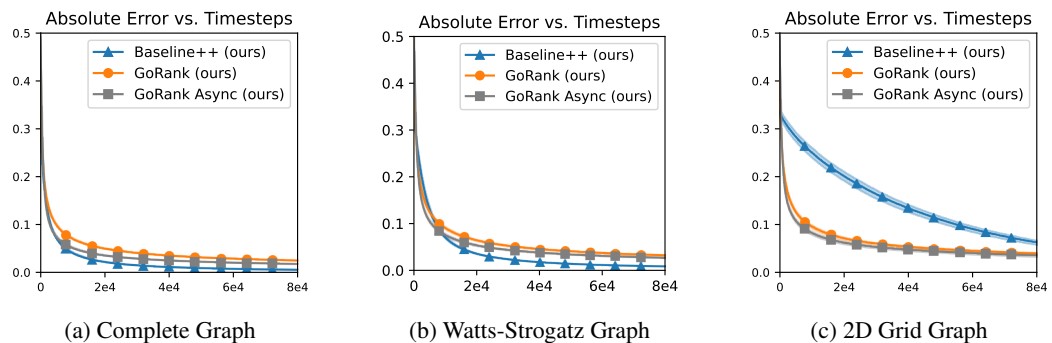

(a) Complete Graph    (b) Watts-Strogatz Graph    (c) 2D Grid Graph

Figure 7: Comparison of Asynchronous GoRank with other ranking algorithms. Results show that Asynchronous GoRank slightly converges faster than the synchronous version.

Note that the convergence analysis in the asynchronous case is more complex because it requires analyzing the ratio of two statistics. Nonetheless, though technically more demanding, it is possible to derive convergence rate bounds using Taylor expansions techniques. We will carry out such an analysis in an extension to this work.

# I   Large-scale Experiments

To demonstrate the scalability of our method on large networks, we repeated the experiments from Fig. (c) in Sections 3.2 and 4.3, originally conducted with $n = 500$, on larger networks with $n = 1000$ and $n = 5000$.

For the experiments related to Section 3.2, the results lead to the same conclusions: Figure 8 shows that GoRank continues to achieve a low error ($< 0.1$), and the overall performance trends of the other ranking algorithms remain similar on the Watts–Strogatz graph but are significantly worse on the 2D grid graph. These additional results further reinforce GoRank's scalability and robustness, particularly in comparison to the other methods.

For the experiments related to Section 5.4, Figure 9 shows that GoTrim + GoRank continues to converge efficiently to the trimmed mean, even on larger 2D grid graphs, clearly outperforming the corrupted mean. In contrast, GoTrim + Baseline++ converges significantly more slowly on the 2D grid (and does not even outperform the corrupted mean), although it remains efficient on Watts–Strogatz graphs.

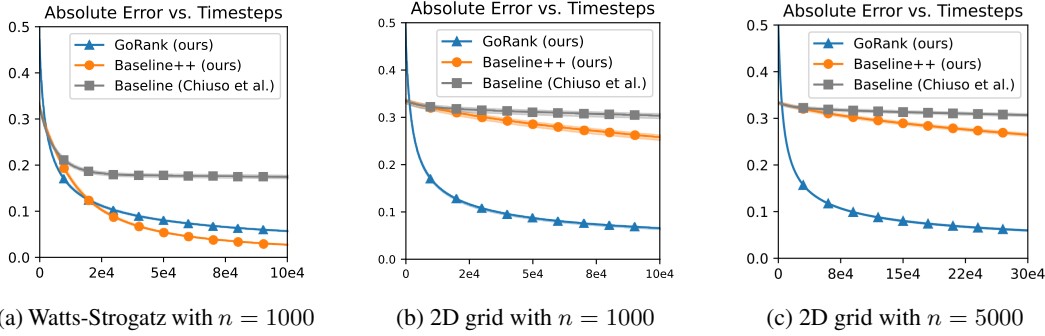

(a) Watts-Strogatz with $n = 1000$    (b) 2D grid with $n = 1000$    (c) 2D grid with $n = 5000$

Figure 8: Comparison of the performance of gossip algorithms for ranking across large networks.

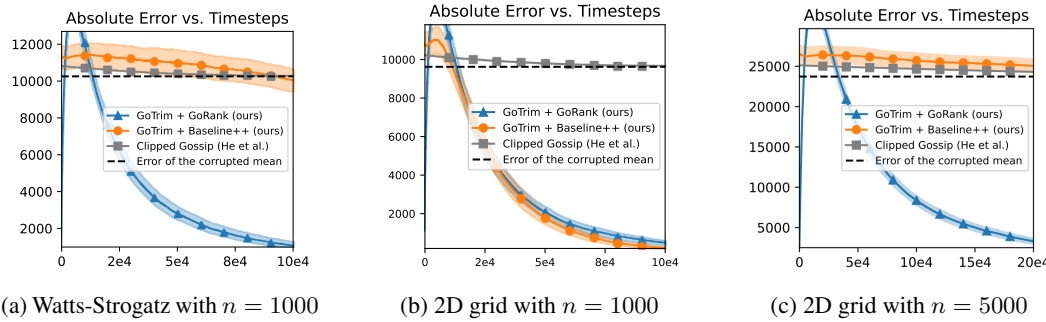

(a) Watts-Strogatz with $n = 1000$    (b) 2D grid with $n = 1000$    (c) 2D grid with $n = 5000$

Figure 9: Comparison of the performance for trimmed means estimation across large networks.

## J Robustness to Network Disruptions

While robustness to data contamination is important, the robustness of our proposed algorithms to network disruptions (e.g., edge/node failures, network partitioning) is equally crucial in real-world applications.

One natural extension is to introduce a fixed probability of failure for each node or edge (see [2]). This would modify the edge sampling: instead of a uniform distribution, the algorithm would sample edge $e$ with probability $p_e$. In our analysis, this change corresponds to replacing the normalized Laplacian $L/|E|$ with a weighted sum $\sum_e p_e L_e$, where $\sum_e p_e < 1$ due to edge failures and $L_e$ correspond to the elementary Laplacians. The spectral gap, which governs the convergence rate, would now correspond to the connectivity constant $\tilde{c}$ of this weighted Laplacian instead of $c = \lambda_2/|E|$. For instance, if each edge fails independently with probability 0.5, then $p_e = 1/2|E|$ for all $e$, and the effective spectral gap becomes $\tilde{c} = \lambda_2/(2|E|) < c$, indicating a slower convergence rate. We emphasize that as long as $p_e > 0$ for all $e$, the weighted graph remains connected and non-bipartite if the original graph was.

Another interesting scenario is network partitioning. One way to model this is by designing graphs composed of tightly connected clusters with only a few inter-cluster edges that are prone to failure. In such a setup, the connectivity constant would degrade significantly, and we expect the convergence rate to reflect this bottleneck. We generated a graph consisting of 500 nodes organized into three well-connected clusters, with only five inter-cluster edges. As expected, we observed a very low connectivity constant $c = 1.86 \times 10^{-6}$ and the convergence behavior of this graph is similar to that of a 2D grid graph, which is consistent with the low overall connectivity.

## K Experiments on Sparse Graphs

An interesting question is whether our algorithms performance well on sparse graphs. First, we note that the Watts-Strogatz and 2D grid graphs used in our main experiments are already quite sparse, with both containing fewer than 1000 edges. However, we would like to emphasize that

what primarily governs convergence behavior is not sparsity per se, but graph connectivity. For example, a dense graph composed of loosely connected clusters may have poor connectivity, while a sparse graph like a 3-regular graph can exhibit strong connectivity properties. A cycle graph, in addition to being sparse, has very low connectivity and serves as a useful pathological case. To better illustrate the relationship between topology and performance, we conducted additional experiments using three different sparse graphs: 1) Watts-Strogatz graph (connectivity $c = 3.31 \times 10^{-4}$, $\tilde{1}000$ edges), 2) Cycle graph ($c = 3.16 \times 10^{-7}$, 499 edges), 3) 3-regular graph ($c = 2.24 \times 10^{-4}$, 750 edges). All experiments were run for $15 \times 10^4$ iterations. For GoRank, both the Watts-Strogatz and 3-regular graphs achieved fast convergence (absolute error below 0.05), consistent with their strong connectivity properties typical of expander graphs. The cycle graph, as expected, performed worse due to its very low connectivity, but still achieved an error below 0.1. For GoTrim, using the same corruption model as in Fig. 5b, the Watts-Strogatz and 3-regular graphs again performed very well, with errors close to 0. However, the cycle graph exhibited much slower convergence: after $8 \times 10^4$ iterations, the absolute error remained above 40. Doubling the number of iterations reduced the error to below 40. While this is indeed slow, it is important to note that performance remains better than the corrupted mean, and such a topology is highly atypical in real-world sensor networks. In practice, such scenarios would likely require gossip algorithms specifically designed for cycle graphs. These experiments support the conclusion that GoTrim still performs well on sparse graph with good connectivity properties.

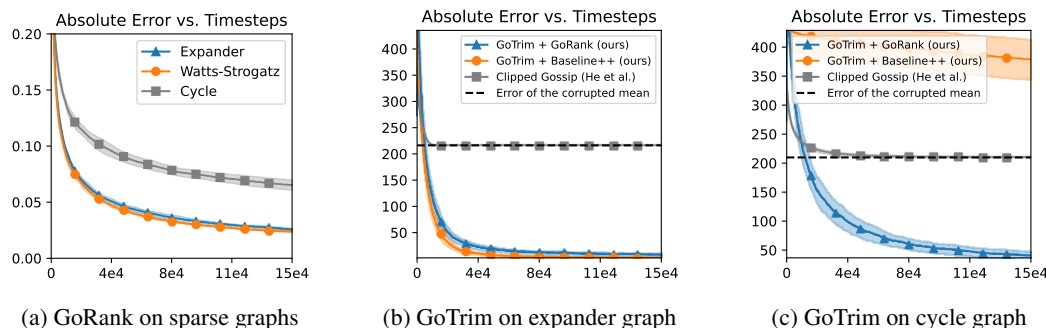

(a) GoRank on sparse graphs  (b) GoTrim on expander graph  (c) GoTrim on cycle graph

Figure 10: Comparison of the performance of GoRank and GoTrim on different sparse graphs.

## L   Further Discussion and Future Work

Our main focus was to develop foundational results for robust decentralized estimation. This section highlights directions for deepening our understanding of the algorithms' properties, extending them to more complex settings, and applying them to related problems.

**Optimality of the Bounds.**   To the best of our knowledge, there are currently no established lower bounds on the convergence rate for the class of gossip-based algorithms applied to either decentralized ranking or trimmed mean estimation. Thus, the optimality of our algorithms remains an open question. In the special case of a complete graph, we observe that significantly faster convergence than the typical $\mathcal{O}(1/t)$ rate is possible. For example, in the Baseline++ algorithm, which performs direct ranking swaps, we can show exponential convergence of the form $\exp(-t/|E|)$. However, this fast convergence critically relies on the high connectivity of the complete graph and does not generalize well to less connected graphs. In contrast, GORANK demonstrates more robust performance across general graphs, including those with limited connectivity, where achieving $\mathcal{O}(1/t)$ convergence is already non-trivial. Deriving a formal lower bound for arbitrary graphs remains an open and challenging problem. Nevertheless, it can be noted that with the current approach, the GORANK convergence rate of $\mathcal{O}(1/t)$ is tight in the sense that rank estimation involves averaging $t$ random indicator functions. Finally, since GOTRIM relies on the rank estimates from GORANK, it inherits the $\mathcal{O}(1/t)$ convergence rate, which is already near-optimal.

**Faster Gossip Algorithms.**   Designing faster gossip algorithms is an interesting direction, and we have also been exploring this aspect. One promising approach involves optimizing the edge sampling strategy based on the graph connectivity constant $c$ (see [4, 38]). While this strategy improves gossip

performance in standard averaging tasks, our empirical experiments suggest that its impact on the GORANK algorithm is more limited. This is likely due to the relatively small variation in $c$, which is insufficient to significantly affect a rate in $1/ct$—though it has a more noticeable effect under a geometric rate.

**Extension to Multivariate Data.**  Extending the proposed approach to multivariate data is an important direction. A natural extension involves ranking multivariate observations based on their norms. Specifically, one could define a suitable norm depending on the task (e.g., Euclidean), compute the norm for each observation, and then apply our univariate ranking method (GoRank) to these one-dimensional values. Observations with the largest norms can then be treated as potential outliers. Following this, a multivariate version of GoTrim can be defined by discarding the top $k = \lfloor \alpha n \rfloor$ observations with the largest norms, and computing the mean of the remaining points using the standard gossip algorithm. Alternatively, one could explore data depths as a generalization of ranking in multivariate settings (see [28]). Data depth provides a measure of centrality for multivariate data. While this approach is promising, it would require a more in-depth investigation beyond the scope of the current work, since depth computations usually require to solve computationally demanding optimization problems, just like alternative methods recently designed to define multivariate ranks (e.g. based on optimal transport). Finally, we could simply compute the coordinate-wise trimmed mean by applying GoTrim to each coordinate individually [29, 39].

**Applications.**  The proposed methods offer a promising foundation for robust decentralized optimization. In particular, GOTRIM could be integrated into existing mean-based optimization algorithms to enhance robustness against outliers [11, 25, 37]. Realizing this integration, however, will require further algorithmic and theoretical development. Additionally, our ranking algorithms may prove valuable in extreme value theory, where identifying rare or extreme observations is critical—especially in high-stakes domains such as finance, insurance, and environmental science [22].

