# OpenReview forum: "Robust Distributed Estimation: Extending Gossip Algorithms to Ranking and Trimmed Means"
_NeurIPS.cc/2025/Conference — NeurIPS 2025 poster_

### Official Review · Reviewer_XdBk · 2025-07-02

**Clarity:** 3
**Significance:** 3
**Originality:** 4
**Rating:** 4
**Confidence:** 3

**Summary:**

The paper studies gossip algorithms for robust mean and rank estimation. In this setting, there exists a graph $G$ over $n$ nodes, as well as a value $X_i$ for each node. For each time step, a uniformly random edge in the graph is chosen, allowing communication of the endpoints of this edge. One should then design an algorithm that, after as few time steps as possible, can find the $\alpha$-trimmed mean of $X$'s for some $0 < \alpha < \frac{1}{2}$, or find the rank of $X_i$ for each node $i$.

The paper establishes the first theoretical results for the convergence rate of such tasks, including an algorithm that achieves $O(1/t)$ convergence rate on rank estimation, and an algorithm that achieves $O(\frac{\log t}{t})$ convergence rate on robust mean estimation, based on the previous rank estimation algorithm. It is worth noting that the hidden factors in the $O$ notations contain the degree of connectivity of $G$.

At a high level, the algorithm for rank estimation works by letting each node $i$ to "see" the labels of all other nodes, done by randomly swapping the labels of the chosen edge at every time step. One should expect that at some time in the future, every label would have walked through every node in the graph, which means every node can now obtain a good estimation of their own rank. For robust mean estimation, one can think of trimming the $\alpha$ outliers as identifying these outliers via finding their ranks, explaining the dependency of robust mean estimation on rank estimation.

Finally, the paper gives numerical experiments showing that their algorithms' performance match the theoretical bound, and are also better than the state-of-the-art.

**Questions:**

**Questions**:
1) Is there any lower bound on the convergence rate achievable via the class of gossip algorithms for the two tasks? It would be nice to understand the gap between the proposed algorithms and the optimal algorithms for the two respective tasks.

**Comments**:
1) In Theorem 3, what is "for all $k \in [n]$" doing in the statement? I don't think the expression below has anything to do with $k$.
2) I think the authors should not output an entire vector for GoTrim, since we're talking about mean estimation here. I believe simply outputting the mean of $Z$ is already good enough due to the L2-norm bound in Theorem 3, and this makes the algorithm a bit less confusing.

**Ethical Concerns:**

["NO or VERY MINOR ethics concerns only"]

**Final Justification:**

I have read the rebuttals and decided to keep my score.

**Limitations:**

yes

**Quality:**

3

**Strengths And Weaknesses:**

**Strengths**:
1) The model is very well motivated while not having a lot of prior work on theoretical bounds, so I think this is a very worthwhile question to study.
2) The algorithms proposed are simple and intuitive, but seem to be better than the current state-of-the-art. In fact, GoRank is the first thing that comes to my mind when reading the model, so I am quite surprised that it performs better than the current SOTA while also satisfying some theoretical bounds.
3) The paper is well-written and seem theoretically sound.

**Weaknesses**:
1) As the authors have mentioned, all of the expected convergence results can be transformed into the high-probability kind using Markov's inequality. However, I believe that a more careful analysis with some standard concentration bounds would benefit this paper a lot, as it enhances the "robustness" aspect of these gossip algorithms.

---

> ### Author Rebuttal · Authors · 2025-07-29
>
> We thank the reviewer for their valuable feedback. We first address the limitation pointed out:
>
> First, let us recall that expected convergence results and high-probability bounds are the standard forms of convergence results in the gossip literature (see [1]). But you are right, this would certainly be interesting to derive concentration bounds in such dependent settings—where each data permutation depends on the previous one—along with the high dimensionality (the state space of permutations is $n!$) and complex data structure (graphs). However this is far from immediate and requires careful analysis. Results such as those in [2] could possibly be applied to establish such bounds. This point will be addressed in the final version of the article, but dealing with it in depth would require an analysis beyond the scope of this work, and we leave it for future research.
>
> We now respond to the two questions raised:
>
> 1. To the best of our knowledge, there are currently no established lower bounds on the convergence rate for the class of gossip-based algorithms applied to either decentralized ranking or trimmed mean estimation.  In the special case of a complete graph, we observe that significantly faster convergence than the typical $\mathcal{O}(1/t)$ rate is possible. For example, in the Baseline++ algorithm, which performs direct ranking swaps, we can show exponential convergence of the form $\exp(-t/|E|)$. However, this fast convergence critically relies on the high connectivity of the complete graph and does not generalize well to less connected graph. In contrast, GoRank demonstrates more robust performance across general graphs, including those with limited connectivity, where achieving $\mathcal{O}(1/t)$ convergence is already non-trivial. Deriving a formal lower bound for arbitrary graphs remains an open and challenging problem. A similar challenge arises in the case of trimmed mean estimation, which depends on accurate rank estimation. We agree that developing tighter theoretical bounds for these problems is an important direction for future work.
>
> 2. The phrase “for all” in Theorem 3 was indeed a typo and should be removed. Regarding your second point, we are not entirely sure we understand your concern, but it seems you are suggesting that the GoTrim algorithm should output only the mean of the vector rather than the full vector itself. First, note that the estimation error we are interested in is whether every node has a good estimator of the trimmed mean of the network. The vector $\boldsymbol{Z}(t)$ is used to capture the error of all the estimators $Z_k(t)$ across the nodes simultaneously, which is a standard formulation in the gossip literature. Importantly, GoTrim outputs an estimate $Z_k(t)$ for each node individually. Would a version of Theorem 3 using the following left-hand term be clearer?
>
> $\frac{1}{n} \sum_{k=1}^n \left|| \mathbb{E}[Z_k(t)] - \bar{x} \right| \leq \frac{1}{\sqrt{n}} \left\| \mathbb{E}[\boldsymbol{Z}(t)] - \bar{x}_\alpha \mathbf{1}_n \right\||_2.$
>
> Or do you suggest the following alternative?
>
> $\left| \mathbb{E}[Z_k(t)] - \bar{x} \right| \leq \|| \mathbb{E}[\boldsymbol{Z}(t)] - \bar{x}_\alpha \mathbf{1}_n \||_2.$
>
> We would appreciate your clarification.
>
> We hope this response has fully addressed your questions and concerns. Please let us know if there is anything else you would like clarified, or any additional information you would need.
>
> [1] Boyd, Stephen, et al. "Randomized gossip algorithms." IEEE transactions on information theory 52.6 (2006).
>
> [2] Bertail, Patrice, and Stéphan Clémençon. "Sharp bounds for the tails of functionals of Markov chains." Theory of Probability \& Its Applications (2010).

---

> > ### Comment · Reviewer_XdBk · 2025-08-04
> >
> > Thank you very much for the detailed response!
> >
> > For point 3, indeed, I have missed that we're in the distributed setting, so congregating every node's estimation and outputting their mean simply does not make sense. I would like to void my original comment on this part, and thank the authors for clearing this up for me.

---

> > > ### Author Response · Authors · 2025-08-04
> > >
> > > Thank you for updating your comment. We're glad the clarification was helpful and hope our response has addressed your remaining concerns as well. Please let us know if there's anything else we can provide to help you consider reevaluating your score.

---

> > > > ### Comment · Reviewer_XdBk · 2025-08-08
> > > >
> > > > Thank you for the response. I intend to keep the score at 4.

---

### Official Review · Reviewer_AiVM · 2025-07-03

**Clarity:** 3
**Significance:** 3
**Originality:** 3
**Rating:** 5
**Confidence:** 2

**Summary:**

This paper proposes two gossip algorithms for robust distributed estimation: GORANK for rank estimation, and GOTRIM for trimmed mean estimation, which builds upon GORANK. The key insight is computing robust statistics by first estimating ranks through pairwise comparisons, then using these ranks to compute weighted averages that exclude outliers. The authors provide theoretical convergence guarantees as well as empirical validation of each approach.

**Questions:**

1. It is noted that the $c^2$ term in the convergence rate of GOTRIM may be overly pessimistic. Can you provide additional ablations on simulated sparse graphs to better illustrate the true performance?
2. Could you comment on the potential applicability of the proposed method to multivariate data?

**Ethical Concerns:**

["NO or VERY MINOR ethics concerns only"]

**Limitations:**

1. The provided bounds may be loose, particularly for sparse graphs. The current work also focuses on simple settings with synchronous communication and Huber faults, rather than more general networks with Byzantine faults.

**Paper Formatting Concerns:**

None.

**Quality:**

3

**Strengths And Weaknesses:**

Strengths:
1. The writing is clear and easy to follow.
2. The method is accompanied by a strong theoretical analysis with convergence rates and breakdown point guarantees.

Weaknesses:
1. Additional comparisons to prior work would be beneficial, both empirically and to provide context at a higher level. A table comparing the convergence rates, requirements on the graph structure, and estimation bias could help clarify the contributions of this work over prior methods.
2. The results are based on a simple data corruption model, rather than considering more general Byzantine faults.

---

> ### Author Rebuttal · Authors · 2025-07-29
>
> We thank the reviewer for their valuable feedback. Below, we address the weaknesses point by point:
>
> 1. While there has been significant work on decentralized estimation of simple aggregates (e.g., average, min, max), to the best of our knowledge, relatively little attention has been given to rank-based statistics in randomized pairwise gossip settings. This is likely due to the inherently global nature of rank-based statistics, which makes them more challenging to estimate in a decentralized setting. As discussed in the related work section (see section 2.2), we identified Chiuso et al. and He et al., which share a similar setting and that can be directly compared to our methods. However, conducting an empirical comparison with other prior works proved to be more difficult. As suggested, we will add a comparative table in the final version that summarizes key aspects of prior methods and our own. This includes whether each method is fully decentralized, whether the estimator is unbiased, and whether theoretical convergence guarantees exist on arbitrary graphs. Such a table could be as follows:
> | **Method**             | **Fully Decentralized?** | **Unbiased?** | **Rates on Any Graphs?** |
> | --------------- | ----------------- | ------------- | ------------------------ |
> | Chiuso et al. (Baseline) [1] | ✗                        | ✓             | ✗                        |
> | Baseline++ (ours)        | ✗                        | ✓             | ✗                        |
> | **GoRank (ours)**        | ✓                        | ✓             | O(1/t)                   |
> | Haeupler et al.  [2]        | ✗                        | ~            | ✗                        |
> | He et al.  [3]              | ✓                        | ✗             | ✗                        |
> | Shrivastava et al.  [4]    | ✗                        | ✓             | ✗                        |
> | **GoTrim (ours)**        | ✓                        | ✓             | O(1/t)                   |
>
> We consider Chiuso et al.’s method not fully decentralized, as it requires an initial global ranking, which may not be available in real-world decentralized settings. Additionally, their convergence analysis is restricted to the complete graph.  Haeupler et al.'s method is also not fully decentralized since it requires sampling $K$ random nodes at the end of the protocol. Furthermore, their algorithm estimates only an approximate median, making the bias unclear. Their theoretical guarantees are again limited to complete graphs. Shrivastava et al.'s method relies on a base station for coordination, and thus does not meet the criteria for full decentralization.
>
> 2. Considering more general Byzantine faults is indeed important. However, we believe that addressing Huber robustness is still relevant and realistic in many practical settings: a sensor could be miscalibrated, stuck at a fixed value, or may consistently report incorrect readings due to environmental factors, without necessarily being attacked.
>
>
> We now respond to the two questions raised:
> 1. First, we note that the Watts-Strogatz and 2D grid graphs used in our main experiments are already quite sparse, with both containing fewer than 1000 edges. However, we would like to emphasize that what primarily governs convergence behavior is not sparsity per se, but graph connectivity. For example, a dense graph composed of loosely connected clusters may have poor connectivity, while a sparse graph like a 3-regular graph can exhibit strong connectivity properties. A cycle graph, in addition to being sparse, has very low connectivity and serves as a useful pathological case. To better illustrate the relationship between topology and performance, we conducted additional ablation studies using three different sparse graphs: 1) Watts-Strogatz graph (connectivity $c = 3.31 \times 10^{-4}$, \~1000 edges), 2) Cycle graph ($c = 3.16 \times 10^{-7}$, 499 edges), 3) 3-regular graph ($c = 2.24 \times 10^{-4}$, 750 edges). All experiments were run for $8 \times 10^4$ iterations. For GoRank, both the Watts-Strogatz and 3-regular graphs achieved fast convergence (absolute error below 0.05), consistent with their strong connectivity properties typical of expander graphs. The cycle graph, as expected, performed worse due to its very low connectivity, but still achieved an error below 0.1. For GoTrim, using the same corruption model as in Fig. 5b, the Watts-Strogatz and 3-regular graphs again performed very well, with errors close to 0. However, the cycle graph exhibited much slower convergence: after $8 \times 10^4$ iterations, the absolute error remained above 40. Doubling the number of iterations reduced the error to below 40. While this is indeed slow, it is important to note that performance remains better than the corrupted mean, and such a topology is highly atypical in real-world sensor networks. In practice, such scenarios would likely require gossip algorithms specifically designed for cycle graphs. These ablations support the conclusion that GoTrim still performs well on sparse graph with good connectivity properties.
>
> 2. We agree that extending the proposed approach to multivariate data is an important direction. A natural extension involves ranking multivariate observations based on their norms. Specifically, one could define a suitable norm depending on the task (e.g., Euclidean), compute the norm for each observation, and then apply our univariate ranking method (GoRank) to these one-dimensional values. Observations with the largest norms can then be treated as potential outliers. Following this, a multivariate version of GoTrim can be defined by discarding the top $k = \lfloor \alpha n \rfloor$ observations with the largest norms, and computing the mean of the remaining points using the standard gossip algorithm. Alternatively, one could explore data depths as a generalization of ranking in multivariate settings (see [5]). Data depth provides a measure of centrality for multivariate data. While this approach is promising, it would require a more in-depth investigation beyond the scope of the current work, since depth computations usually require to solve computationally demanding optimization problems, just like alternative methods recently designed to define multivariate ranks (e.g. based on optimal transport). Finally, we could simply compute the coordinate-wise trimmed mean by applying GoTrim to each coordinate individually. A discussion about such possible extensions will be added in a dedicated section of the Supplementary Material.
>
> Finally, regarding limitations, we note that an asynchronous extension of GoRank has been presented, which also enables an asynchronous version of GoTrim.
>
> We hope this response has fully addressed your questions and concerns. Please let us know if there is anything else you would like clarified, or any additional information you would need.
>
> [1] Chiuso, Alessandro, et al. "Gossip algorithms for distributed ranking." Proceedings of the 2011 American Control Conference. IEEE (2011).
>
> [2] Haeupler, Bernhard, Jeet Mohapatra, and Hsin-Hao Su. "Optimal gossip algorithms for exact and approximate quantile computations." Proceedings of the 2018 ACM Symposium on Principles of Distributed Computing (2018).
>
> [3] He, Lie, Sai Praneeth Karimireddy, and Martin Jaggi. "Byzantine-robust decentralized learning via clippedgossip." arXiv preprint arXiv:2202.01545 (2022).
>
> [4] Shrivastava, Nisheeth, et al. "Medians and beyond: new aggregation techniques for sensor networks." Proceedings of the 2nd international conference on Embedded networked sensor systems (2004).
>
> [5] Mosler, Karl. "Depth statistics." Robustness and complex data structures: Festschrift in Honour of Ursula Gather (2013).

---

> > ### Comment · Reviewer_AiVM · 2025-08-08
> >
> > Thank you for the detailed response. While I understand the challenges in comparing methods with differing problem settings and assumptions, the draft comparison table still feels incomplete. Including convergence rates for all methods (perhaps with a separate column for the complete graph setting) would strengthen the presentation. Overall, my concerns have been addressed, so I will maintain my original score.

---

> > > ### Author Response · Authors · 2025-08-09
> > >
> > > We thank you for your feedback and are glad our response has addressed your concerns. We agree that adding a column with the convergence rates for the complete graph setting would improve the table. We will update the table accordingly in the final version of the article.

---

### Official Review · Reviewer_eSuy · 2025-07-03

**Clarity:** 3
**Significance:** 2
**Originality:** 2
**Rating:** 4
**Confidence:** 3

**Summary:**

The paper introduces two decentralized gossip algorithms designed to robustly estimate ranks and trimmed means in networks with corrupted or malicious nodes. The authors provide a theoretical convergence analysis for both methods, which is insightful, and experiments validating the theoretical results on diverse network topologies and data contamination scenarios.

**Questions:**

- Can you give a more clear motivation on why rank estimation, and trimmed mean estimation in the gossip setting is practically relevant. Including why it's important over the whole graph as opposed to taken locally for each neighborhood of a node?
- Does the method extend to random-edge gossip (sampling one edge at a time)? This would be a pragmatic way to go towards asynchronous?
- Currently some clarity is missing on Byzantine vs weaker notions of robustness. In the discussion of He et all ClippedGossip, is it fair to me to use fixed clipping radius? Could you clarify the high-level results comparison of types of robustness (Byzantine vs just robust) achieved/not achieved.

**Ethical Concerns:**

["NO or VERY MINOR ethics concerns only"]

**Final Justification:**

I'm fine with either outcome on this paper. The rebuttal clarified minor aspects so I'm not having major concerns remaining. The real-world example they provided is only half convincing though: the method does not give privacy guarantees, and if that's the case then centralized methods are applicable as well and scale rather well as we're exchanging scalars only, not LLM-sized model parameters

**Limitations:**

See above

**Paper Formatting Concerns:**

-

**Quality:**

3

**Strengths And Weaknesses:**

Strengths:
- Novelty of extension, and solid mathematical formulation of the problems.
- Breakdown point analysis is insightful

Weaknesses:
- The comparison to Byzantine robustness is left unclear. What exactly is the threat model? (Discussing practical relevance also)

---

> ### Author Rebuttal · Authors · 2025-07-29
>
> We thank the reviewer for their valuable feedback. Below, we respond to the four questions raised:
>
> 1. In this work, we adopt the model of honest nodes with corrupted data, following Huber's contamination model, in which a fraction of the data consists of outliers. The nodes perform updates correctly and consistently based on their local observations but their local observation could be an outlier. This distinguishes our setting from the Byzantine model, where nodes may behave arbitrarily or maliciously, potentially sending incorrect updates with the intent to disrupt consensus or degrade performance. To clarify this distinction, we propose to add the following remark in Section 2.1 explicitly comparing Huber robustness with Byzantine robustness:
>
> "Remark 2. Our setup assumes honest nodes, meaning they perform updates correctly and consistently based on their local observations. However, under Huber’s contamination model, these observations may include outliers. Note that this setup is fundamentally different from the Byzantine model where nodes may behave arbitrarily or maliciously, potentially sending incorrect updates with the intent to disrupt consensus or degrade performance."
>
> We underline that our model is still realistic in many practical settings: a sensor could be miscalibrated, stuck at a fixed value, or may consistently report incorrect readings due to environmental factors, without necessarily being attacked. Although less challenging, Huber robustness is still an important problem that needed to be addressed.
>
> 2. We clarify the motivation as follows:
>
> a) Rank estimation over the whole graph is crucial because it provides a measure of how extreme a particular value is in a global context. A node that is a global outlier may appear normal when compared only to its immediate neighbors, particularly if those neighbors are also outliers. In gossip-based algorithms, where values propagate across the network, even a single undetected extreme value can significantly skew global statistics such as the mean. Therefore, identifying and quantifying outliers requires comparisons across the entire graph, not just local neighborhoods.
>
> b) Trimmed mean estimation likewise requires a global perspective. A locally computed trimmed mean (e.g., using values from a node’s neighborhood) may still be biased or corrupted if a majority of neighboring nodes are outliers. For such an algorithm to be reliable, we would need strong assumptions about each node’s neighborhood, which is often unrealistic, particularly in poorly connected graphs. We have explored deterministic gossip algorithms that use the median rule as the aggregator, meaning that at each iteration, each node computes the median of its neighbors. While this approach does converge to a consensus, our experiments showed that it is biased (especially when the graph is sparse/poorly connected) and does not converge to the true median.
>
> Overall, both rank and trimmed mean are inherently global statistics—unlike local aggregates such as averages, minima, or maxima, which can rely on local conservation principles to ensure convergence to the correct global values. No such local conservation principle exists for ranks or trimmed means, which justifies the need for our proposed approach.
>
> 3. Yes, our method does indeed extend naturally to a fully asynchronous setting, where only two nodes (one edge) update at each iteration. We addressed this extension (see Appendix H), where we introduce an asynchronous version of GoRank. In this variant, each node maintains a local counter $C_k$ that tracks the number of updates it has participated in, replacing the use of a global iteration counter $t$. The update rule becomes $(1 - \frac{1}{C_k}) R_k' + \frac{1}{C_k} \mathbb{I}_{X_k > Y_k}$. Empirically, we observed that Asynchronous GoRank converges slightly faster than its synchronous counterpart, and is more efficient. Furthermore, note that GoTrim inherits the asynchronous nature of GoRank when used in this setting. Since this is an interesting extension, we have added the following remark in section 3.1 discussing it in detail:
>
> "Remark 3. The asynchronous extension of GoRank is straightforward: it replaces the global iteration counter $t$ with a local counter $C_k$ maintained at each node $k$. The update rule then becomes $(1 - \frac{1}{C_k}) R_k' + \frac{1}{C_k} \mathbb{I}_{X_k > Y_k}.$ Empirically, we find that Asynchronous GoRank converges slightly faster and is more efficient than its synchronous counterpart (see Appendix H)."
>
> However, in this paper, we focus primarily on the synchronous setting to simplify the theoretical analysis. The asynchronous case is more complex because it requires analyzing the ratio of two statistics. Nonetheless, though technically more demanding, it is possible to derive convergence rate bounds using Taylor expansions techniques. We have carried out such an analysis in an extension to this work. For completeness, we may refer to it.
>
> 4. Please see our response to Question 1 for a more detailed discussion of the distinction between Byzantine robustness and weaker notions of robustness.
> Regarding the use of a fixed clipping radius in our analysis of ClippedGossip: yes, it is fair to use a fixed clipping radius, as this is consistent with the experimental setup in He et al.’s original work. Although He et al. mention an adaptive clipping mechanism, its applicability to a randomized pairwise gossip setting is unclear. Specifically, adaptive clipping typically relies on statistics computed over neighbors' data, which may not be readily available in a pairwise setting. Moreover, such a scheme appears to require a well-connected network topology and introduces an additional hyperparameter, which limits its practicality in our setup. As noted in our response to Question 1, this work specifically focuses on robustness to data corruption under the assumption that all nodes are honest. In contrast, Byzantine robustness is a fundamentally different setting. While Byzantine robustness is indeed an important direction, it falls outside the scope of this paper. In the final version we will write that the development of a rigorous theoretical framework for Byzantine robustness remains an open problem in this field, which we will address in the future.
>
> We hope this response has fully addressed your questions and concerns. Please let us know if there is anything else you would like clarified, or any additional information you would need.

---

### Official Review · Reviewer_VJLP · 2025-07-07

**Clarity:** 3
**Significance:** 3
**Originality:** 3
**Rating:** 4
**Confidence:** 5

**Summary:**

The paper introduces two novel gossip algorithms designed for robust distributed estimation over arbitrary communication graphs: GORANK for rank estimation and GOTRIM for trimmed mean estimation. The authors demonstrate that GORANK provides efficient rank estimation with an $\mathcal{O}(1/t)$ convergence rate, while GOTRIM achieves robust trimmed mean estimation with an $\mathcal{O}(\log(t)/t)$ convergence rate. Both algorithms are analyzed in terms of their theoretical convergence rates and breakdown point analysis, with empirical validation also provided. The proposed algorithms aim to improve the robustness of distributed learning in decentralized networks, particularly in the presence of corrupted or adversarial nodes.

**Questions:**

- Is there any comparison of GOTRIM with other trimmed mean estimation techniques (like those based on median or quantile estimations) in terms of both theoretical bounds and practical performance?
- Though GORANK is quite intuitive, $t$ in Algorithm 1 Step 4 seems to be abused. I wonder if the weights of $R_k^{\prime}$ and $\mathbb{I}_{X_k > Y_k}$ is varying across different iterations?

**Ethical Concerns:**

["NO or VERY MINOR ethics concerns only"]

**Final Justification:**

I will keep my positive score for potential acceptance.

**Limitations:**

See 'Weakness' part.

**Paper Formatting Concerns:**

No Formatting Concern

**Quality:**

3

**Strengths And Weaknesses:**

Strengths
- The proposed GORANK and GOTRIM algorithms extend gossip-based methods in a meaningful way by incorporating robust ranking and trimmed mean estimation. This is a significant contribution to decentralized, robust statistical estimation.
- GOTRIM’s breakdown point analysis further demonstrates the robustness of the trimmed mean estimate even in the presence of corrupted data.
- The experiments are extensive, covering different graph topologies and contamination schemes, and show that both algorithms outperform existing methods in terms of robustness to outliers and performance in poorly connected graphs.

Weaknesses
- The experiments conducted involve relatively small network sizes (e.g., 500 nodes), which may not fully represent the scalability of the proposed methods. The performance of the algorithms in larger networks or at scale has not been sufficiently tested.
- The experiments assume that the network graph is connected and non-bipartite. However, in real-world applications, graphs might experience node failures, network partitioning, or other disruptions. It would be useful to see robustness tests under these conditions.
- While the authors provide upper bounds for the convergence rates of GORANK and GOTRIM, there is no discussion on whether these rates are tight. The paper does not explore the possibility of faster gossip-based algorithms or variance reduction strategies that might improve the convergence rates.
- Some parts of the paper are notation-heavy, and symbols like $\sigma_{n}(r_{k}),  c_{n}(r_{k})$, and $\gamma_{k}$ are introduced quickly without much surrounding context or intuition. Readers not deeply familiar with decentralized estimation literature may find it difficult to parse the implications of these quantities.
- The paper provides upper bounds on the convergence rates of GORANK and GOTRIM but does not discuss whether these bounds are optimal or how they compare to known lower bounds in decentralized or robust estimation (if any). This leaves open the question of whether the proposed methods are theoretically optimal.

---

> ### Author Rebuttal · Authors · 2025-07-29
>
> We thank the reviewer for their valuable feedback. Below, we address the weaknesses point by point:
>
> 1. It is indeed important to demonstrate the scalability of our method on larger networks. The experiments from Fig (c) in Section 3.2 and 4.3 conducted with n=500 were also carried out with larger networks of sizes of $n = 1000$ (using $t = 10^5$ iterations) and $n = 5000$ (using $t = 3\cdot 10^5$ iterations). Regarding the experiments related to Section 3.2, the results lead to the same conclusions: GoRank continues to achieve a low error (< 0.1), and the overall performance trends of the other ranking algorithms remain similar, although they are generally slower as the network size increases. These additional results further reinforce GoRank's scalability and robustness, particularly in comparison to the other methods. Regarding the experiments related to Section 5.4, we observed that GoTrim + GoRank continues to converge efficiently to the trimmed mean, even on larger 2D grid graphs. In contrast, GoTrim + Baseline++ converges significantly more slowly on the 2D grid, although it remains efficient on Watts-Strogatz graphs. These additional experiments will be included in the Supplementary Material of the revised version for the sake of completeness. Scalability issues on very large networks (e.g. $n>100 000$) are beyond the scope of this paper.
>
> 2. The robustness of our proposed algorithms to network disruptions is indeed an important consideration in real-world applications. First, note that randomized gossip algorithms (our setup) are inherently more robust to node/edge failures than deterministic algorithms, since they operate by sampling a single edge at each iteration and do not require full connectivity at every step (see [1]). Moreover, since our synchronous setup requires all nodes to update at each iteration, node failures could be a problem. However this is easily solved with the asynchronous setup we proposed in the appendix, since in this setting, each node keeps a local counter and is not affected by a missing update. That said, we agree that explicitly incorporating failures into our framework would enhance its practical relevance. One natural extension is to introduce a fixed probability of failure for each node or edge (see [2]). This would modify the edge sampling: instead of a uniform distribution, the algorithm would sample edge $e$ with probability $p_e$. In our analysis, this change corresponds to replacing the normalized Laplacian $L/|E|$ with a weighted sum $\sum_e p_e L_e$, where $\sum_e p_e < 1$ due to edge failures and $L_e$ correspond to the elementary Laplacians. The spectral gap, which governs the convergence rate, would now correspond to the connectivity constant $\tilde{c}$ of this weighted Laplacian instead of $c=\lambda_2/|E|$. For instance, if each edge fails independently with probability 0.5, then $p_e = 1/2|E|$ for all $e$, and the effective spectral gap becomes $\tilde{c} = \lambda_2 / (2|E|) < c$, indicating a slower convergence rate. We emphasize that as long as $p_e > 0$ for all $e$, the weighted graph remains connected and non-bipartite if the original graph was.  Regarding network partitioning, this is indeed an interesting direction to explore. One way to model this is by designing graphs composed of tightly connected clusters with only a few inter-cluster edges that are prone to failure. In such a setup, the connectivity constant would degrade significantly, and we expect the convergence rate to reflect this bottleneck. We generated a graph consisting of 500 nodes organized into three well-connected clusters, with only five inter-cluster edges. As expected, we observed a very low connectivity constant $c = 1.86 \times 10^{-6}$. Empirically, the convergence behavior of this graph is similar to that of a 2D grid graph, which is consistent with the low overall connectivity of the graph structure. The experiments mentioned will be included in the Supplementary Material in an additional section dedicated to robustness to topology changes.
>
> 3. Regarding the tightness of the bounds, please see point 5 below. Designing faster gossip algorithms is indeed an interesting direction, and we have also been exploring this aspect. One promising approach involves optimizing the edge sampling strategy based on the graph connectivity constant $c$ (see [3]). While this strategy improves gossip performance in standard averaging tasks, our empirical experiments suggest that its impact on the GoRank algorithm is more limited. This is likely due to the relatively small variation in $c$, which is insufficient to significantly affect a rate in $1/ct$—though it has a more noticeable effect under a geometric rate. Regarding variance reduction strategies, we agree that they could potentially enhance convergence. However, the application of these techniques would require stronger assumptions and a more extensive theoretical framework. While these questions will be discussed in the final version of the article, providing answers to them is beyond the scope of the current paper.
>
>
> 4. We agree that the notation in some parts of the paper could be simplified. The symbols $\sigma_{n}(r_{k})$, $c_{n}(r_{k})$, and $\gamma_{k}$ are constants that arise naturally in the derivation of our convergence bounds. Their expressions are somewhat complex, so we used simplified notation to keep the results readable. The dependence on $n$ and $r_k$ was included for completeness, but we understand it may add unnecessary complexity without providing much intuition. These constants are specific to our analysis and do not come from the standard decentralized estimation literature. In the revision, we will simplify the notation—for example, replacing $\sigma_{n}(r_k)$ with $\sigma_k$—to improve clarity.
>
> 5. We agree that a discussion on the tightness of the convergence bounds would strengthen the theoretical presentation. To the best of our knowledge, there are currently no established lower bounds for the specific problems addressed in our work, the optimality of our algorithms remains an open question. For GoRank, the convergence rate of $O(1/t)$ is indeed tight in the sense that rank estimation involves averaging $t$ random indicator functions. As a result, the convergence is necessarily in $O(1/t)$. However, the constant $1/c$ is not tight in the bound due to the use of an upper bound of a finite geometric sum. This simplification was useful for obtaining interpretable expressions. For GoTrim, we noticed that the original bound was actually not optimal and included an unnecessary $\log t$ term. Upon closer analysis, we realized that in the proof of Theorem 3, we used a crude bound of $\sum_{s=T+1}^{t} \lambda^{t-s} \leq t - T$, leading to the $\log t$ dependence. However, this term can be more tightly bounded by $1/c$, and we have updated the proof of Theorem 3 accordingly in the camera-ready version. Finally, since GoTrim relies on the rank estimates from GoRank, it inherits the $O(1/t)$ convergence rate, which is already near-optimal. While Boyd et al. have shown that gossip algorithms for averaging have optimal convergence rates (see [1]), this approach does not directly apply to our setting due to the different nature of the problem. We agree that deriving formal general lower bounds is an important question and this will be the subject of further research.
>
>
> We now respond to the two questions raised:
>
> 1. To the best of our knowledge, there are no existing quantile or median estimation methods that are directly applicable to our setting---specifically, a randomized pairwise gossip framework operating over arbitrary graph topologies. That said, our GoRank algorithm can be adapted for median estimation by propagating the observation whose current rank estimate is close enough to the rank of the median and most recent.
> We have also explored gossip-based algorithms that use a median update rule. One variant we tested involves using three neighboring nodes and updating their estimate to the median of the three values. Another variant maintains a "wandering observation" $Y_k$ for each node $k$, and upon communication between nodes $i$ and $j$, updates $Z_i$ and $Z_j$ using the median of $(Z_i, Y_i, Y_j)$ and $(Z_j, Y_j, Y_i)$ respectively, while swapping $Y_i$ and $Y_j$. Despite converging to a consensus, these algorithms had a strong bias and did not converge to the true median on average in our empirical evaluations.
>
> 2. You are correct that the weights do vary across iterations, and we realize this may not have been sufficiently clear in the original version. In Algorithm 1, the variable $t$ denotes the current iteration counter. For example, at $t = 1$, the weight on $R_k^{\prime}$ is $w_1 = 0$ and the weight on $\mathbb{I}_{X_k > Y_k}$ is $w_2 = 1$; at $t = 2$, we have $w_1 = \frac{1}{2}$ and $w_2 = \frac{1}{2}$; and so on. This effectively performs a running average. We emphasize that this implementation assumes a \emph{synchronous} setting, where all nodes share the same iteration counter $t$. For the \emph{asynchronous} case (as discussed in the Appendix), $t$ can be replaced by a local counter $c_k(t)$ for each node $k$. To avoid confusion between $t$ as an iteration counter and $t$ as a time horizon, we now use $s$ in Algorithms 1 and 2 to denote the iteration counter.
>
> We hope this response has fully addressed your questions and concerns. Please let us know if there is anything else you would like clarified, or any additional information you would need.
>
> [1] Boyd, Stephen, et al. "Randomized gossip algorithms." IEEE transactions on information theory 52.6 (2006).
>
> [2] Haeupler, Bernhard, Jeet Mohapatra, and Hsin-Hao Su. "Optimal gossip algorithms for exact and approximate quantile computations." Proceedings of the 2018 ACM Symposium on Principles of Distributed Computing (2018).
>
> [3] Boyd, Stephen, Persi Diaconis, and Lin Xiao. "Fastest mixing Markov chain on a graph." SIAM review 46.4 (2004).

---

> > ### Comment · Reviewer_VJLP · 2025-08-05
> > **Official Comment by Reviewer VJLP**
> >
> > Thanks for the rebuttal, and I will keep my positive score for potential acceptance.

---

### Note · Authors · 2025-08-12

We thank the reviewers for their insightful comments and suggestions. In our opinion, we have been able to address all of the concerns raised. The main clarifications and improvements we have made are summarized below:
- **Motivation for robustness to outliers (Huber's model):** We clarified the motivation for Huber robustness with a concrete real-world example, and distinguished Huber robustness from Byzantine robustness through a new remark.
- **Asynchronous extensions:** We clarified that an asynchronous extension is already provided in the appendix, while the main text focuses on the synchronous setting for clarity of the theoretical analysis. We also noted that the asynchronous analysis has been carried out in an extension of this work and may be referenced for completeness.
- **Extension to network disruptions:** We provided a remark on extending our theoretical framework to handle network disruptions (e.g., node failures, network partitioning).
- **Notations:** We suggested to simplify several notations for clarity.
- **Related works:** We provided a summary table assessing whether each method is fully decentralized, whether the estimator is unbiased, and whether theoretical convergence rates exist (on complete and arbitrary graphs).
- **Approach:** We explained that a global approach is necessary because our statistics are global quantities, requiring a different strategy from classical gossip algorithms.
- **New experiments:** We referred to additional experiments to demonstrate scalability on larger networks and strong performance on well-connected sparse graphs.
- **Discussion:** We provided a discussion on faster gossip algorithms, the tightness of bounds, and potential extensions to multivariate data.

Moreover, according to the reviewers, this paper provides:
- a "significant contribution to decentralized, robust statistical estimation" with "a well-motivated model";
- a "solid mathematical formulation of the problems", "strong theoretical analysis with convergence rates", and an "insightful" breakdown point analysis;
- "extensive" experiments, "covering different graph topologies and contamination schemes" which shows that "both algorithms outperform existing methods".

The reviewers also mentioned that the paper was "well-written",  "clear", and "easy to follow".

In light of these positive evaluations and the improvements made, we hope our contribution reaches NeurIPS's standards.

---

### Decision · Program_Chairs · 2025-09-17

**Decision:**

Accept (poster)

**Comment:**

This paper proposes two gossip-based algorithms, GoRank and GoTrim, for robust distributed estimation under Huber’s contamination model. The theoretical analysis is solid, with convergence and breakdown guarantees, and the experiments show improvements over existing methods. Reviewers appreciated the novelty, clarity of the mathematics, and the practical validation. Concerns remain about scalability, the relation to Byzantine robustness, and the tightness of the bounds, though the rebuttal addressed some of these points. Overall, the contribution is significant and the consensus leans positive, so I recommend acceptance.